# DRIVE: Distributional Model-Based Reinforcement Learning via Variational Inference

## Abstract

Distributional reinforcement learning (RL) provides a natural framework for estimating the distribution of returns rather than a single expected value. However, the control aspect of distributional RL has not been as thoroughly explored as the evaluation part, typically relying on the greedy selection rule with respect to either the expected value, akin to standard approaches, or risk-sensitive measures derived from the return distribution. On the other hand, casting RL as a probabilistic inference problem allows for flexible control solutions utilizing a toolbox of approximate inference techniques; however, its connection to distributional RL remains underexplored. In this paper, we bridge this gap by proposing a variational approach for efficient policy search. Our method leverages the log-likelihood of optimality as a learning proxy, decoupling it from traditional value functions. This learning proxy incorporates aleatoric uncertainty of the return distribution, enabling risk-aware decision-making. We provide a theoretical analysis of our framework, detailing the conditions for convergence. Empirical results on vision-based tasks in DMControl Suite demonstrate the effectiveness of our approach compared to various algorithms, as well as its ability to balance exploration and exploitation at different training stages.

## 1 Introduction

The return, composed of cumulative rewards, is a central component of RL, summarizing how effective an agent is. Standard RL (Sutton & Barto, 2018) aims to maximize the expected value of returns to improve the agent's decisions. While this approach is widely adopted in the literature, it ignores the underlying distributional nature of the returns rooted in the randomness of transitions. For example, two returns with the same expected value can exhibit different levels of variability. In such cases, standard RL fails to distinguish between them. In contrast, distributional RL (Bellemare et al., 2017) directly models the distribution of returns, allowing for the incorporation of aleatoric uncertainty. For instance, a risk-averse agent would prefer lower variance, while a risk-seeking agent might tolerate higher variance. A substantial body of works (Dabney et al., 2018b) (Dabney et al., 2018a) (Yang et al., 2019) focus on improving the approximation quality of such distributions based on the distributional Bellman operator (Bellemare et al., 2017). However, with regard to the control aspect – specifically, how to refine the policy in relation to the return distribution for risk-aware decision making, existing research is limited. Most approaches derive a statistic from the return distribution, either the expectation or risk-sensitive measures, to greedily improve the policy. This raises the question: can we develop a new control principle that better aligns with the nature of distributional RL beyond the current scope?

Control as probabilistic inference (Levine, 2018) provides a promising framework for our purpose. This framework represents the underlying dynamical system using a probabilistic graphical model (PGM) and associates the rewards with an additional *optimality variable*. Conventionally, this optimality variable is often proportional to the exponential rewards. This choice has been shown to link the maximization of the log-likelihood to that of cumulative rewards (Toussaint, 2009), thereby connecting the probabilistic inference with RL. Its application has been demonstrated in previous literature from various angles. For instance, one can match to the posterior after observing the optimality variables (Rawlik et al., 2013) or maximize the likelihood of a trajectory being optimal

(Abdolmaleki et al., 2018). Moreover, through the lens of message passing (Pearl, 1982) or KL divergence minimization (Rawlik et al., 2013), these formulations can give rise to several categories of algorithms, including those in Maximum Entropy RL (Ziebart, 2010) or variational policy search (Neumann, 2011) (Peters & Schaal, 2007) (Hachiya et al., 2009) (Abdolmaleki et al., 2018). Furthermore, probabilistic methods such as expectation maximization, expectation propagation (Minka, 2001), or recent advancements in variational inference (Kingma & Welling, 2014), can be effectively utilized by those algorithms. However, despite being versatile, interpretable, and powerful, the application of probabilistic inference to distributional RL remains underexplored, even when the return variable can be readily incorporated into the graphical model. To bridge this gap, we aim to explore how to model the control aspect of distributional RL within the probabilistic inference framework and uncover the insights this new approach would bring.

In this paper, we introduce DRIVE, a distributional model-based RL algorithm designed for efficient policy search through variational inference. We develop probabilistic learning proxies as alternatives to traditional value functions, transforming the standard RL problem into a distributional framework. The return variable is incorporated into this framework by encoding information about the return distribution into the optimality variable through marginalization. We leverage the variational inference to jointly optimize a practical variational lower bound, iteratively improving the desired objective. Since approximating our objective involves sampling trajectories from a model, we integrate our method with model-based approaches like Dreamer (Hafner et al., 2020) to learn a transition model. Theoretical analysis is conducted to understand convergence and the optimization process. Empirical results demonstrate the effectiveness of our approach on challenging vision-based tasks in DMControl Suite, enhancing the uncertainty-aware decision-making.

## 2 PRELIMINARIES

We consider an infinite-horizon discounted Markov Decision Process $(\mathcal{S}, \mathcal{A}, P, R, \rho_0, \gamma)$, where $\mathcal{S}$ and $\mathcal{A}$ represent the state and action spaces, $P$ the transition kernel $P(\cdot|s, a)$, $R$ the reward function, $\rho_0$ the initial state distribution, and $\gamma \in [0, 1)$ the discount factor. This process models how the agent interacts with the environment. At each step, the agent takes an action $a_t \sim \pi(\cdot|s_t)$ at the current state $s_t$, and receives a reward $R(s_t, a_t)$, and transits to a new state $s_{t+1} \sim P(\cdot|s_t, a_t)$. Following this procedure, we can define the return as $U^\pi(s_t, a_t) = \sum_{k=0}^{\infty} \gamma^k R(s_{t+k}, a_{t+k})$, which is a random variable. Whenever noted, we denote the approximate transition model as $\hat{f}$. We assume the reward function is bounded, therefore the return is also bounded. We denote the maximum of the return as $U_{\max}$. The action value function is defined as $Q^\pi(s, a) = \mathbb{E}[U^\pi(s, a)]$, characterized by:

$$Q^\pi(s, a) = \mathbb{E}[R(s, a)] + \gamma \mathbb{E}_{P,\pi}[Q^\pi(s', a')]. \tag{1}$$

The value function is then the expected value of action value function, $V^\pi(s) = \mathbb{E}_\pi[Q^\pi(s, a)]$.

This approach succinctly represents the agent's objective in terms of the expectation; however, it is unable to capture the underlying distributional information, as the dynamics, reward function, or policy could be stochastic.

### 2.1 DISTRIBUTIONAL REINFORCEMENT LEARNING

In contrast, distributional RL (Bellemare et al., 2017) directly models the distribution of the return instead of a single expected value. In this perspective, the distributional Bellman operator is defined as:

$$\mathcal{T}^\pi U(s, a) \overset{\mathrm{D}}{:=} R(s, a) + \gamma U(s', a') \qquad s' \sim P(\cdot|s, a), a' \sim \pi(\cdot|s')$$

$$(\mathcal{T}^\pi)^H \underbrace{U(s_t, a_t)}_{p(U|s,a)} \overset{\mathrm{D}}{:=} \underbrace{R_{<H} + \gamma^H U(s_{t+H}, a_{t+H})}_{\mathsf{q}(U|s,a)} \qquad \tau \sim P, \pi, \ R_{<H} := \sum_{n=0}^{H-1} \gamma^n R(s_{t+n}, a_{t+n}),$$

$$\tag{2}$$

where the equality denotes two random variables have equal probability laws, and $\tau$ is the trajectory generated under the transition model $P$ and the policy $\pi$. We denote the distribution of $U(s, a)$ as $p(U|s, a)$ and $(\mathcal{T}^\pi)^H U(s, a)$ as $\mathsf{q}(U|s, a)$, which is the bootstrapped return distribution, derived by expanding the one-step operator $H - 1$ times.

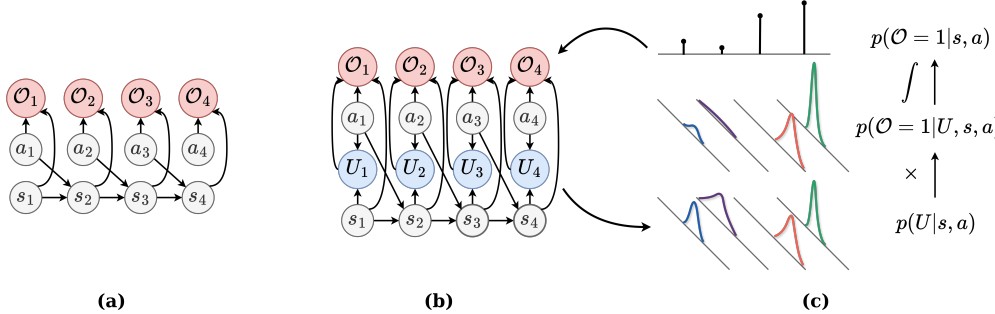

(a)          (b)          (c)

Figure 1: Comparison of PGM between the standard approach and our method: **(a)** Optimality variables are embedded and conditioned on state and action; **(b)** Our method first incorporates the return variable $U$, which then conditions the optimality variables; **(c)** Procedure overview: (i) establish a prior on $p(\mathcal{O} = 1|U, s, a)$ (ii) marginalize the product of the return distribution and this prior to obtain the conditional optimality distribution.

While most previous works have focused on improving the approximation of the return distribution based on the distributional Bellman operator, little attention has been paid to improving the policy based on the return distribution. They typically follow a greedy selection rule, where the improved policy $\pi'$ corresponds to $\max_{a \in \mathcal{A}} \mathbb{E}[U^\pi(s, a)] = Q^\pi(s, a)$. However, this is similar to the standard RL and discards the uncertainty information within the return distribution for decision-making. Our goal is to use probabilistic inference to incorporate this information.

### 2.2 RL as Probabilistic Inference

To embed the control problem into a graphical model, we need to introduce a binary random variable $\mathcal{O}$ which denotes optimal if $\mathcal{O} = 1$ otherwise it is not optimal. Typically, this variable is related to an exponential transformation on the reward (Todorov, 2008) (Rawlik et al., 2013), (Levine, 2018):

$$p(\mathcal{O} = 1|s, a) \propto \exp(R(s, a)). \tag{3}$$

This formulation results in a steeper curve as the reward increases. It bears a close relationship to energy-based methods (Haarnoja et al., 2017) and Maximum Entropy RL algorithms (Haarnoja et al., 2018) by minimizing the KL divergence between the trajectory distribution and the posterior after observing optimality variables. Other derivatives generally adhere to this principle although the interpretation of the optimality variable may differ, for instance, in the finite-horizon case[1], the likelihood of a trajectory being optimal is:

$$p(\mathcal{O} = 1|\tau) \propto \exp\left(\sum_{t=0}^{T} R(s_t, a_t)\right). \tag{4}$$

However, these formulations have their own shortcomings. The first approach is limited to individual steps, failing to account for cumulative information. While the second approach addresses this limitation by considering past events, it does not capture environmental uncertainty. To resolve these issues, we propose a new formulation that not only considers future events but also captures uncertainty. Figure 1 illustrates a comparison between the standard approach and our method.

## 3 CONTROL AS INFERENCE

### 3.1 FROM STANDARD RL TO DISTRIBUTIONAL PERSPECTIVE

First of all, we propose a probabilistic learning proxy that allows us to transfer from the standard RL formulation to the distributional setting.

---

[1]Extending to the infinite horizon case simply needs follow a modified dynamic $\bar{P}(\cdot|s, a) = \gamma P(\cdot|s, a) + (1 - \gamma)\delta(s = \bar{s})$ where $\bar{s}$ is an absorbing state regardless of what action has been taken.

The goal of the standard RL is to find an optimal policy such that $\pi^\star(\cdot|s) = \arg\max_\pi V^\pi(s)$ for all states $s \in \mathcal{S}$. Instead, we consider maximizing a probabilistic learning proxy that represents the log-likelihood of being optimal. Notably, it can be related to the corresponding state-action counterpart in a manner analogous to how the value function is expressed as the expectation of the action value function:

$$\max_\pi V^\pi(s) = \mathbb{E}_\pi[Q^\pi(s,a)], \forall s \in \mathcal{S} \tag{5}$$

$$\max_\pi \log p^\pi(\mathcal{O}=1|s) = \log \mathbb{E}_\pi[p^\pi(\mathcal{O}=1|s,a)], \forall s \in \mathcal{S}. \tag{6}$$

This formulation offers a more natural framework for probabilistic inference by decoupling the optimization problem from traditional value functions. However, adapting to distributional RL raises the question of how to holistically integrate the return with this probabilistic learning proxy.

## 3.2 VARIATIONAL BOUND

To address this problem, we integrate the return $U$ into the state-action probabilistic learning proxy by marginalizing over all possible outcomes of the return distribution. We then employ the concept of variational inference to infer the most probable action distributions based on that probabilistic learning proxy. Thereafter, we decompose the objective by associating it with the bootstrapped return distribution $\mathfrak{q}(U|s,a)$. This approach fosters: 1) long-horizon policy optimization, 2) divergence-awareness in return distribution predictions, and 3) direct balancing of the exploration-exploitation trade-off with an appropriate model specification.

In the first step, we model aleatoric uncertainty in $U$ using a parametric return model $p_\psi(U|s,a)$ and a likelihood model $p(\mathcal{O}=1|U,s,a)$. By marginalizing over $U$, we can incorporate this uncertainty into the state-action probabilistic learning proxy:

$$\log p_\psi(\mathcal{O}=1|s,a) = \log \int p(\mathcal{O}=1|U,s,a)p_\psi(U|s,a)dU. \tag{7}$$

Different choices for the likelihood model can lead to varying agent behaviors. In this paper, we define our model as being proportional to the exponential of $U$:

$$p(\mathcal{O}=1|U,s,a) \propto \exp(U). \tag{8}$$

With this model specification, we find that it can effectively balance the exploration and exploitation trade-off.

Next, we utilize variational inference to solve the problem in Equation 6. To facilitate a tractable approximation, we make the following assumption:

**Assumption 3.1.** $p(\mathcal{O}=1|U_{\max},s,a)^2 = 1$.

It is worth noting that Assumption 3.1 is easy to validate as we assume the reward function is bounded.

Based on our model specification in Equation 8 and Assumption 3.1 regarding $p(\mathcal{O}=1|U,s,a)$, we derive a variational lower bound using Jensen's inequality. The policy, value distribution, and variational posterior are parameterized as $(\theta, \psi, \phi)$, respectively, where the variational posterior $q_\phi(a|\mathcal{O}=1,s)$ approximates the true posterior:

$$\log p_\psi^{\pi_\theta}(\mathcal{O}=1|s) \geq -D_{\mathrm{KL}}(q_\phi(a|\mathcal{O}=1,s)||\pi_\theta(a|s))$$

$$+ \mathbb{E}_{q_\phi(a|\mathcal{O}=1,s)}[\log \int \underbrace{p(\mathcal{O}=1|U,s,a)}_{\propto \exp(U)} p_\psi(U|s,a)dU]$$

$$\geq - \underbrace{D_{\mathrm{KL}}(q_\phi(a|\mathcal{O}=1,s)||\pi_\theta(a|s))}_{\mathcal{J}_{\mathrm{KL}}^{(1)}} \tag{9}$$

$$+ \underbrace{\mathbb{E}_{q_\phi(a|\mathcal{O}=1,s),\mathfrak{q}(U|s,a)}[U]}_{\mathcal{J}_U}$$

$$- \underbrace{\mathbb{E}_{q_\phi(a|\mathcal{O}=1,s)}[D_{\mathrm{KL}}(\mathfrak{q}(U|s,a)||p_\psi(U|s,a))]}_{\mathcal{J}_{\mathrm{KL}}^{(2)}} - U_{\max}$$

$$:= \mathcal{L}(\theta,\phi,\psi;s),$$

[2]$U_{\max}$ can be relaxed as $U_{\max} + \epsilon$ as long as $\epsilon \geq 0$.

The overall objective comprises three terms: complexity $\mathcal{J}_{\text{KL}}^{(1)}$, reparameterized policy gradient (PG) $\mathcal{J}_U$, and regularizer $\mathcal{J}_{\text{KL}}^{(2)}$. This structure offers multiple benefits. The complexity term facilitates policy optimization through two models – the policy and the variational posterior – by dividing the multi-step optimization problem into two manageable parts. Additionally, the reparameterized PG term enables long-horizon optimization via importance weighting with the bootstrapped return distribution, allowing for more information from the future to be backpropagated into both the policy and the variational posterior. Moreover, the regularizer measures the discrepancy between the return distribution and the bootstrapped return distribution. For actions with a significant discrepancy, the variational posterior will reduce the likelihood of those actions, which then influences the policy through the complexity term, fostering divergence-aware decision-making.

In the next section, we focus on how to approximate those terms with a practical transition model $\hat{f}$.

## 3.3 DECOMPOSITION

**Complexity $\mathcal{J}_{\text{KL}}^{(1)}$**  The complexity term is generally tractable with simple distributions, such as Gaussian or Beta, but requires approximation with complex distributions, like Gaussian mixtures.

**Reparameterized PG $\mathcal{J}_U$**  By definition of $\mathfrak{q}(U|s,a)$ and leveraging the change of variables, we can expand $\mathcal{J}_U$ over multiple steps:

$$\mathcal{J}_U = \mathbb{E}_{q_\phi,\pi_\theta,\hat{f},p_\psi(U|s_{t+H},a_{t+H})} \left[ R_{<H} + \gamma^H U(s_{t+H}, a_{t+H}) \right], \tag{10}$$

which intuitively recovers the discounted cumulative rewards. Additionally, it can be efficiently optimized using Monte Carlo estimates when all components are reparameterized.

**Regularizer $\mathcal{J}_{\text{KL}}^{(2)}$**  The approximation of the regularizer reduces to approximating $\mathfrak{q}(U|s,a)$. If the return distribution belongs to the Normal distribution class, it can be expressed analytically as a weighted combination of Normal distributions based on the trajectory distribution:

$$\mathfrak{q}(U|s,a) = \mathbb{E}_{\pi_\theta,\hat{f}} \left[ \mathcal{N}(R_{<H} + \gamma^H \mu_\psi(s_{t+H}, a_{t+H}), \gamma^{2H} \sigma_\psi^2(s_{t+H}, a_{t+H})) \right]. \tag{11}$$

Furthermore, we can derive statistics that are useful for approximating $\mathfrak{q}(U|s,a)$:

$$\begin{aligned}
\mathbb{E}[U|s,a] &= \mathbb{E}_{\pi_\theta,\hat{f}}[R_{<H} + \gamma^H \mu_\psi(s_{t+H}, a_{t+H})] \\
\text{Var}[U|s,a] &= \gamma^{2H} \mathbb{E}_{\pi_\theta,\hat{f}}[\sigma_\psi^2(s_{t+H}, a_{t+H})] + \text{Var}_{\pi_\theta,\hat{f}}[R_{<H} + \gamma^H \mu_\psi(s_{t+H}, a_{t+H})].
\end{aligned} \tag{12}$$

In practice, we can generate $N$ trajectories for each $(s,a)$ and then empirically estimate those quantities to approximate $\mathfrak{q}(U|s,a)$ for calculating the regularizer $\mathcal{J}_{\text{KL}}^{(2)}$.

# 4 PRACTICAL ALGORITHM

In this section, we outline our final objectives and demonstrate how to integrate them with a model-based approach for long-horizon prediction.

An intriguing property of our variational lower bound is that it unifies policy and value distribution updates into a single objective. By differentiating it with respect to $\psi$, we obtain the cross-entropy loss for the value distribution based on the target distribution $\mathfrak{q}(U|s,a)$. Additionally, by differentiating it with respect to both $\theta$ and $\phi$, we can jointly optimize the posterior and the policy.

$$\texttt{Value Dist:}\ \mathcal{J}(\psi) = \mathbb{E}_{\mathfrak{q}(U|s,a)}[-\log p_\psi(U|s,a)] \tag{13}$$

$$\texttt{Posterior + Policy:}\ \mathcal{J}(\theta,\phi) = -\mathcal{J}_U + \mathcal{J}_{\text{KL}}^{(1)} + \mathcal{J}_{\text{KL}}^{(2)}. \tag{14}$$

In order to approximate $\mathcal{J}_U$, we need a transition model $\hat{f}$ to sample trajectories. We opt the RSSM of Dreamer (Hafner et al., 2020) to enable long-horizon prediction. With a deterministic encoder $h_t = \text{GRU}(h_{t-1}, s_{t-1}, a_{t-1})$ tracking the history information, the overall generative model will be:

$$\begin{aligned}
\text{Representation model:} &\quad \mathfrak{q}(s_t|h_t, o_t) \\
\text{Observation model:} &\quad p(o_t|h_t, s_t) \\
\text{Reward model:} &\quad p(r_t|h_t, s_t) \\
\text{Transition model:} &\quad p(s_t|h_t).
\end{aligned} \tag{15}$$

These terms can be jointly optimized by improving a variational lower bound across multiple time steps:

$$\mathcal{J}_{\text{Dreamer}} = \sum_{t=1}^{T} \mathbb{E}_{\text{q}}[\underbrace{\log p(o_t \mid h_t, s_t)}_{\mathcal{J}_{\text{O}}^t} + \underbrace{\log p(r_t \mid h_t, s_t)}_{\mathcal{J}_{\text{R}}^t} - \underbrace{D_{\text{KL}}(\text{q}(s_t \mid h_t, o_t) \parallel p(s_t \mid h_t))}_{\mathcal{J}_{\text{KL}}^t}]. \quad (16)$$

In summary, our algorithm DRIVE encompasses three phases – data collection, model learning, and behavior learning. A key aspect of behavior learning in DRIVE is that it branches at the first time step, dividing the multi-step optimization problem into two manageable parts handled by the posterior and the policy. This not only amortizes the policy optimization but also allows for efficient optimization via stochastic gradient methods. A full procedure of behavior learning is outlined in Algorithm 1.

---

**Algorithm 1** DRIVE: Behavior Learning

---

**Denote** $x_t = (h_t, s_t)$
**Initialize parameters** $\phi, \theta, \psi$
> Behavior Learning
Imagine $H$-length trajectories $\{(x_\tau, a_\tau)\}_{\tau=t}^{t+H}$ from each $x_t$ with $a_t \sim q_\phi(\cdot | \mathcal{O} = 1, x_t)$ otherwise $a_\tau \sim \pi_\theta(\cdot | x_\tau), \tau > t$.
Sample rewards $r_{t+\tau} \sim p(r_{t+\tau} | x_{t+\tau}), \tau = 0, 1, \ldots, H-1$.
Sample values $U_{t+H} \sim p_\psi(U_{t+H} | x_{t+H}, a_{t+H})$.
Compute $H$-step return as targets $\widehat{U}_t$ for each $(x_t, a_t)$.
Estimate $\text{q}(U_t | x_t, a_t)$ with rewards $r_{t+\tau}$ and statistics $(\mu_{t+H}, \sigma_{t+H})$ (Equation 12).
Update posterior and policy (Equation 14).
Update value distribution with $\widehat{U}_t$ (Equation 13).

---

## 5 THEORETICAL ANALYSIS

In this section, we present a theoretical analysis of our method. Its complexity stems from involving not only a changing prior (the policy) but also a truncated optimization with a finite horizon $H$. This contrasts with standard approximate inference methods like VAE or EM, where the prior is typically fixed. Additionally, unlike in RL, the probabilistic decoder in these methods does not depend on the $H$-step value distribution or value function. Given these challenges, our analysis aims to identify conditions under which our method would converge, ideally to a local optimum.

Let us consider a two-stage problem interleaving between the optimization of the approximate posterior $q$ and the policy $\pi$ as follows:

$$\begin{aligned}
\mathcal{J}(q, \pi) &= -D_{\text{KL}}(q \| \pi) + \mathbb{E}_q[\log p^\pi(\mathcal{O} = 1 | s, a)] \\
&= -D_{\text{KL}}(q \| \pi) + \mathbb{E}_q[\log p_H^\pi(\mathcal{O} = 1 | s, a, \pi)],
\end{aligned} \quad (17)$$

where we made use of a shorthand $\log p_H^\pi(\mathcal{O} = 1 | s, a, \pi)$ such that when $\tilde{\pi}$ equates $\pi$ in what follows:

$$\log p_H^\pi(\mathcal{O} = 1 | s_t, a_t, \tilde{\pi}) := \log \mathbb{E}_{\tilde{\pi}, P, p^\pi(U | s_{t+H}, a_{t+H})} \left[ \exp \left( R_{<H} + \gamma^H U \right) \right] - U_{\max}. \quad (18)$$

Optimizing $\mathcal{J}(q, \pi)$ can be divided into two subproblems: $(a)$ $\max_q \mathcal{J}(q, \pi)$ and $(b)$ $\max_\pi \mathcal{J}(q^\pi, \pi)$, where $q^\pi$ is the optimum of problem $(a)$. Notably, for the problem $(b)$, not merely can $\pi$ approach to $q^\pi$ but also be optimized within $\log p_H^\pi(\mathcal{O} = 1 | s, a, \pi)$ for a fixed $H$-step horizon.

As will be shown, the repeated two-stage step will produce a monotonic policy sequence that at least converges to a local optimum $\pi^\star$ under some conditions to account for the bias of the value distribution in $\log p^\pi(\mathcal{O} = 1 | s, a, \tilde{\pi})$.

Define $g^\pi(s, a) := \mathbb{E}_{p^\pi(U | s, a)}[\exp(\gamma^H U)]$, we obtain:

**Theorem 5.1.** *For a given initial policy $\pi_0$, the two-state optimization, if satisfying:*

$$\mathbb{E}_{q^{\pi_k}} \left[ \log \frac{\mathbb{E}_{\tau | \pi_{k+1}, P} \left[ \exp \left( R_{<H} \right) g^{\pi_{k+1}}(s_{t+H}, a_{t+H}) \right]}{\mathbb{E}_{\tau | \pi_{k+1}, P} \left[ \exp \left( R_{<H} \right) g^{\pi_k}(s_{t+H}, a_{t+H}) \right]} \right] \geq 0 \quad (19)$$

*produces a monotonic improving sequence of policies $\{\pi_k\}$ such that*

$$\log p^{\pi_{k+1}}(\mathcal{O} = 1 | s) \geq \log p^{\pi_k}(\mathcal{O} = 1 | s), \quad (20)$$

*which converges to a local optimum $\pi^\star$ such that:*

$$\lim_{k\to\infty} \log p^{\pi_k}(\mathcal{O}=1|s) = \log p^{\pi^\star}(\mathcal{O}=1|s) \geq V^{\pi^\star}(s) - U_{max}. \tag{21}$$

However in practice, directly calculating $\log p^\pi(\mathcal{O}=1|s,a)$ poses challenges in both numerical stability and expectation approximation. Specifically, 1) exponential intensifies large returns, potentially leading to overflow; 2) multiple trajectories are required to approximate the expectation, which can be inefficient. Alternatively, we could trade off the accuracy with improved stability and sample efficiency by utilizing the following surrogate:

$$\begin{aligned}
\mathcal{L}(q,\pi) &= -D_{\mathrm{KL}}(q||\pi) + \mathbb{E}_{q,p^\pi(U|s,a)}[U] \\
&= -D_{\mathrm{KL}}(q||\pi) + \mathbb{E}_q[Q_H^\pi(\pi)],
\end{aligned} \tag{22}$$

where similar to Equation 18, we have $Q_H^\pi(\pi)$ as follows:

$$Q_H^\pi(s_t, a_t; \tilde{\pi}) := \mathbb{E}_{s_{t+1}, a_{t+1}\sim\tilde{\pi}, \cdots, s_{t+H}, a_{t+H}\sim\tilde{\pi}} \left[ \sum_{k=0}^{H-1} \gamma^k R(s_{t+k}, a_{t+k}) + \gamma^H Q^\pi(s_{t+H}, a_{t+H}). \right]. \tag{23}$$

This is akin to SVG($\infty$) (Heess et al., 2015) on finite-horizon trajectories, or reparameterized PG in our context by modifying the distribution to which expectations adhere while preserving the action value function under the original policy $\pi$ at the final time step.

**Theorem 5.2.** *For a given initial policy $\pi_0$, the two-state optimization over surrogate $\mathcal{L}(q,\pi)$, if satisfying:*

$$\mathbb{E}_{\substack{q^{\pi_k}(a_t|\mathcal{O}=1,s_t), \\ P(s_{t+H}|s_t,a_t), \\ \pi_{k+1}(a_{t+H}|s_{t+H})}} [Q^{\pi_{k+1}}(s_{t+H}, a_{t+H}) - Q^{\pi_k}(s_{t+H}, a_{t+H})] \geq 0 \tag{24}$$

*produces a monotonic improving sequence of policies $\{\pi_k\}$ such that:*

$$\log \mathbb{E}_{\pi_{k+1}}[\exp Q^{\pi_{k+1}}] \geq \log \mathbb{E}_{\pi_k}[\exp Q^{\pi_k}] \tag{25}$$

*which converges to a local optimum $\pi^\star$ such that:*

$$\lim_{k\to\infty} \log \mathbb{E}_{\pi_k}[\exp Q^{\pi_k}] = \log \mathbb{E}_{\pi^\star}[\exp Q^{\pi^\star}] \geq V^{\pi^\star}(s). \tag{26}$$

# 6 EXPERIMENTS

In this section, we aim to understand the effectiveness and advantages of DRIVE. We evaluate DRIVE on diverse and challenging continuous control tasks from DMControl Suite (Tassa et al., 2018), including tasks with high-dimensional state and action spaces, dense and sparse rewards, and image observations. We seek to answer the following questions:

- **(1)** How does DRIVE compare with model-based, distributional RL, and "RL as inference" approaches?
- **(2)** Does DRIVE effectively balance the exploration and exploitation during training?
- **(3)** What are the roles of different components of DRIVE's objective?

**Baselines** We evaluation our method against the following:
— **Dreamer and its successors**, the base model (Hafner et al., 2020) used in our approach, which is a state-of-the-art model-based approach enabling long-horizon prediction. Successive developments have improved not only the model learning but also the control aspect, including mixed actor gradients, entropy regularization (Hafner et al., 2021) and advantage normalization (Hafner et al., 2023).
— **TD-MPC** (Hansen et al., 2022), another model-based approach, integrates model predictive control to achieve sample-efficient control.
— **D4PG** (Barth-Maron et al., 2018), an adaption of distribution RL for continuous control, derived from DDPG.
— **SAC** (Haarnoja et al., 2018), an off-policy RL algorithm closely tied to probabilistic inference, whose objective aligns with matching the trajectory distribution to the posterior (Levine, 2018).

Table 1: Evaluation on vision-based DMControl Suite. We report the mean and 95% confidence interval of the average return across 5 random seeds, each with 1M frames. The results of prior methods are sourced from either official reports or open-source repositories. $\sim$ indicates the results are estimated based on the results from the original paper.

| Tasks | Dreamer | DreamerV2 | DreamerV3 | SAC pixel | TD-MPC ($\sim$) | MPO state ($\sim$) | D4PG pixel (100M) | DRIVE |
|---|---|---|---|---|---|---|---|---|
| Ball in Cup Catch | $967 \pm 4$ | $797 \pm 291$ | $\mathbf{972 \pm 5}$ | $173 \pm 92$ | $\mathbf{973}$ | $970$ | $981 \pm 1$ | $962 \pm 16$ |
| Cheetah Run | $716 \pm 32$ | $741 \pm 67$ | $\mathbf{777 \pm 45}$ | $25 \pm 14$ | $583$ | $675$ | $524 \pm 7$ | $767 \pm 60$ |
| Finger Spin | $517 \pm 179$ | $397 \pm 58$ | $\mathbf{791 \pm 125}$ | $269 \pm 59$ | $\mathbf{990}$ | $975$ | $986 \pm 1$ | $647 \pm 182$ |
| Finger Turn Easy | $777 \pm 63$ | $891 \pm 32$ | $834 \pm 115$ | $141 \pm 67$ | $725$ | $950$ | $971 \pm 4$ | $\mathbf{907 \pm 77}$ |
| Finger Turn Hard | $716 \pm 111$ | $842 \pm 63$ | $\mathbf{896 \pm 85}$ | $79 \pm 81$ | $500$ | $840$ | $966 \pm 3$ | $872 \pm 65$ |
| Quadruped Run | $389 \pm 64$ | $490 \pm 80$ | $371 \pm 53$ | $59 \pm 39$ | $388$ | $-$ | $-$ | $\mathbf{648 \pm 77}$ |
| Quadruped Walk | $444 \pm 63$ | $\mathbf{719 \pm 80}$ | $474 \pm 137$ | $79 \pm 25$ | $425$ | $-$ | $-$ | $670 \pm 263$ |
| Reacher Easy | $610 \pm 112$ | $959 \pm 8$ | $933 \pm 42$ | $77 \pm 34$ | $738$ | $975$ | $967 \pm 4$ | $\mathbf{977 \pm 12}$ |
| Walker Run | $\mathbf{720 \pm 37}$ | $684 \pm 78$ | $775 \pm 15$ | $29 \pm 5$ | $606$ | $825$ | $567 \pm 19$ | $654 \pm 59$ |
| Walker Stand | $957 \pm 10$ | $969 \pm 4$ | $\mathbf{983 \pm 8}$ | $139 \pm 24$ | $965$ | $980$ | $985 \pm 1$ | $982 \pm 17$ |
| Walker Walk | $956 \pm 10$ | $959 \pm 1$ | $962 \pm 12$ | $37 \pm 11$ | $960$ | $970$ | $968 \pm 2$ | $\mathbf{974 \pm 15}$ |
| Task Mean | $685$ | $766$ | $797$ | $101$ | $714$ | $-$ | $-$ | $\mathbf{815}$ |

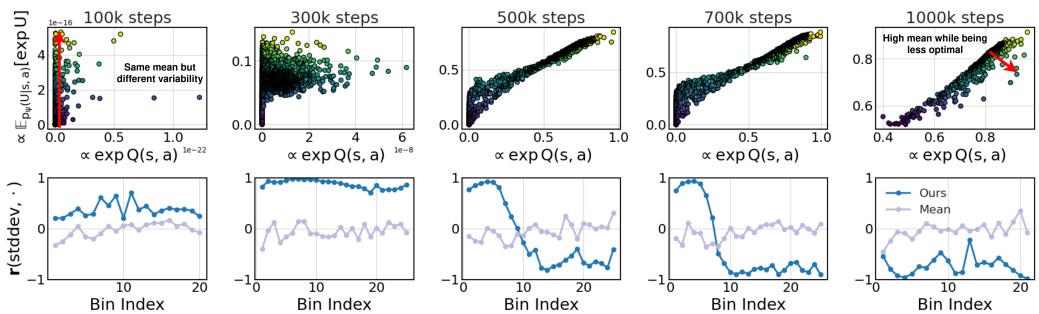

Figure 2: Comparison of optimality criterion under the return distribution and its mean across training. **Top**: The dispersion of our criterion with respect to the mean in ascending order; **Bottom**: The correlation between stddev of the return distribution and two criteria on evenly spaced bins.

— **MPO** (Abdolmaleki et al., 2018), a variational policy search algorithm combined with expectation maximization, shares similarities with (Levine & Koltun, 2013), where a variational lower bound on the log-likelihood of optimality is utilized.

The overall results are shown in Table 1. They demonstrate that our approach is competitive with or outperforms other methods on most tasks, including model-based, distributional RL, and "RL as inference" approaches. Specifically, improved sample-efficiency can be observed in our approach compared to distributional RL with greedy selection rule. Furthermore, when inference is combined with distributional RL, it shows advantages over previous "RL as inference" algorithms.

**Balancing Exploration and Exploitation**   One problem regarding standard approaches is that relying on a single expected value overlooks the uncertainty inherent in the return distribution. This issue becomes particularly significant when either the policy, dynamics or reward function is stochastic. Consequently, we monitor how our optimality criterion varies with respect to the mean of the return distribution (transformed by exponential) throughout the learning process. As illustrated in Figure 2, two key observations emerge: **(1)** Return distributions with the same mean value are not necessarily equally optimal according to our criterion; **(2)** A higher mean may be less optimal. This indicates that, beyond the mean value, the variability within the distribution also affects optimality. To explore how this variability influences optimality, we calculate the correlation coefficient $\mathbf{r}(\text{stddev}, \cdot)$ between the standard deviation of the return distribution (stddev) and the two criteria. From Figure 2, we observe that, in the early stages of training, our criterion is positively correlated with stddev, encouraging exploration. However, this correlation becomes more negative as the policy becomes more optimal, shifting the focus toward exploitation. This demonstrates that our method effectively balances exploration and exploitation at different stages of training, improving the uncertainty-aware decision-making. In contrast, the mean shows a consistent near-zero correlation with the variability in the return distribution, which complicates the handling of novel situations.

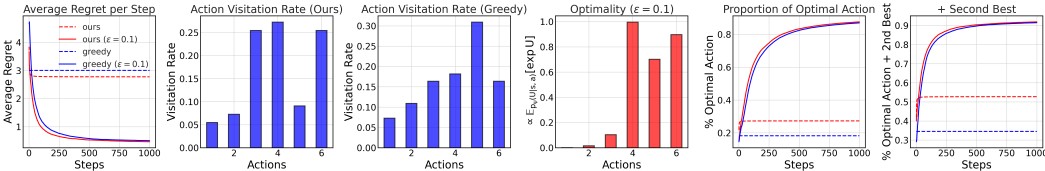

Figure 3: A 6-armed truncated Normal bandit with aleatoric uncertainty. The optimal action is arm 4, which is distracted by arm 5 with the same mean but higher uncertainty. Mean of 50 seeds.

Additionally, we test our method on a 6-armed bandit problem in the presence of aleatoric uncertainty (Figure 3). Compared to the greedy policy, our method not only effectively explores actions with mediocre expected value but high variability, but also exploits the optimal action by avoiding high uncertainty. In contrast, the greedy policy often gets stuck in a suboptimal action and fails to sufficiently explore other promising actions. For more details, please refer to the Appendix C.3.

**Disentanglement** Our objective offers several key benefits. Firstly, the variational posterior divides the multi-step policy optimization into two manageable parts by branching at the first time step. Meanwhile, the regularizer term assesses the quality of the return distribution, penalizing actions with a significant discrepancy to the bootstrapped return distribution. We investigate the roles of these two terms by replacing the posterior with the policy and removing the regularizer term, disentangling their influences on overall policy optimization. As shown in Figure 4(a), this leads to respective performance degradation, validating benefits of both the variational posterior and the regularizer. In addition, we examine the effect of varying the number of trajectories per data point generated from the world model for approximating the terms in our objective. From Figure 4(b), we find that increasing the number of trajectories negatively impacts performance, with $N = 1$ typically being sufficient. One hypothesis we propose to explain this phenomenon is that a greater number of generated trajectories increases the likelihood of exploiting model errors in unreliable predictions. Furthermore, regarding computational complexity, we do not observe significant overhead from the presence of the posterior network and the new objective, as shown in Table 4(c).

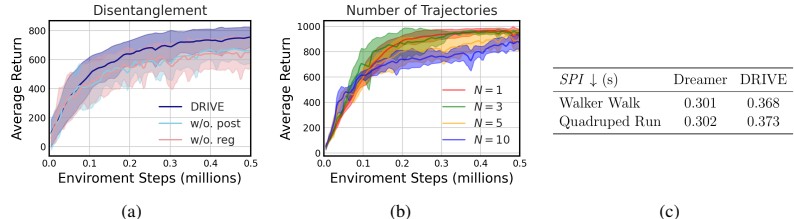

| $SPI \downarrow$ (s) | Dreamer | DRIVE |
|---|---|---|
| Walker Walk | 0.301 | 0.368 |
| Quadruped Run | 0.302 | 0.373 |

(a)          (b)          (c)

Figure 4: **(a)** Disentanglement of policy and posterior, as well as the effect of regularizer, aggregated across 5 tasks; **(b)** Different numbers of generated trajectories from world model; **(c)** Second per iteration (SPI), time required to complete one iteration of both *model learning* and *behavior learning*.

## 7 RELATED WORK

**Distributional RL** While the distributional perspective of RL has been explored since early times (Jaquette, 1973) (Sobel, 1982) (White, 1988), it has gained systematic attention more recently through (Bellemare et al., 2017). This approach has shown promising results on discrete domains with parametric quantile (Dabney et al., 2018b), implicit return distribution (Dabney et al., 2018a), or mixed between (Yang et al., 2019). For continuous domains, different solutions were developed, including Gaussian mixture models (Nam et al., 2021), extension upon DDPG (Barth-Maron et al., 2018) (Lillicrap et al., 2016), generative modeling (Yue et al., 2020), and sample-based approaches (Singh et al., 2022) (Shahriari et al., 2022). Even more, its application in robotic applications (Schneider et al., 2023) showcased risk-sensitive behaviors. However, one major concern is that while the evaluation part has seen consistent improvement, exploration of the control aspect has

been less fruitful. Originally, the policy used was based entirely on the mean of the return distribution (Bellemare et al., 2017), just as in standard RL. This principle persisted until (Dabney et al., 2018a) pointed out this limitation, advocating for the use of distortion risk measures to adjust the distribution under which the expectation obeys. In contrast, our approach adopts the perspective from probabilistic inference, enabling uncertainty-aware decision making.

**Control under Risk**    "Risk" refers to the uncertainty over possible outcomes (Dabney et al., 2018a). In this regard, control under risk is about how to handle this uncertainty. Typically, a risk-neutral agent would only wish to maximize the expected return, without considering any variability within the distribution. However, with pessimistic or optimistic estimates, it can be classified as risk-averse or risk-seeking, respectively. Various approaches exist to induce these behaviors by controlling a single risk parameter, such as free-energy (Howard & Matheson, 1972), cumulative probability weighting (Tversky & Kahneman, 1992) expected shortfall (Rockafellar et al., 2000), and distortion operators (Wang, 2000). While most of those methods focus on finding a distortion risk measure, our approach is more closely related to expected utility theory (Von Neumann & Morgenstern, 1947), where a functional transformation is applied to the return without alternating its distribution. We believe our method has potentials to incorporate various types of functional transformations beyond the exponential.

**RL as Inference**    Probabilistic inference has a rich history in RL. Early works often focused on optimizing open-loop action sequences using methods like EM algorithm (Dayan & Hinton, 1997) or maximum a posteriori (Attias, 2003). Conversely, connecting "costs" with probabilities can be traced back to optimal control methods, such as Kalman duality (Todorov, 2008), KL divergence control (Rawlik et al., 2013), and trajectory optimization (Toussaint, 2009). On the other hand, RL relates this probability to "rewards" to enhance the policy search for reward transformation (Peters & Schaal, 2007), multiple situations (Neumann, 2011), efficient exploration (Ziebart, 2010) (Levine & Koltun, 2013), sample-efficiency (Abdolmaleki et al., 2018), and solving POMDPs (Toussaint et al., 2006). Recent advancements in RL with deep learning have further expanded those concepts from various perspectives, such as energy-based policy (Haarnoja et al., 2017) and soft policy iteration (Haarnoja et al., 2018). Additionally, (Levine, 2018) provided a unified view of those methods within the framework of probabilistic inference. Framing RL as an inference problem offers benefits from the rich toolbox of inference techniques, including parametric or non-parametric approaches and efficient approximate inference methods, which enhance expressiveness, interpretation, and reasoning among nodes. However, extending this framework to distributional RL remains untapped. Our approach therefore effectively bridges this gap.

**Model-based RL**    Model-based RL aims to learn a transition model from experiences, which is beneficial for planning as it eliminates the need to interact with the environment directly. This approach has demonstrated higher sample efficiency by utilizing synthetic data (Sutton, 1990), improved value estimates (Feinberg et al., 2018), and multi-step planning (Oh et al., 2017). However, in practice, as model errors accumulate, the predictions can become less reliable (Janner et al., 2019), especially in high-dimensional spaces and under partial observability. To mitigate these challenges, learning the dynamics in a compact latent space (Hafner et al., 2019) has emerged as a more efficient approach, which enables long-horizon prediction and multi-task learning. However, while much attention has been focused on improving this representation, relatively little has been devoted to policy optimization. Typical approaches involve reparameterized PG with $\lambda$-return (Sutton, 1988). Our approach can be seen as an exploration in this direction, providing alternative ways for efficient policy search.

## 8    CONCLUSION

In this paper, we proposed a methodology bridging the gap between distributional RL and probabilistic inference regarding the control aspect. Our contribution lies in probabilistic learning proxies in place of traditional value functions and a variational inference objective. When combined with model-based approaches, a distributional model-based RL algorithm – DRIVE is derived. Theoretical analysis offers insights into the conditions for convergence and the optimization behaviors. Empirical results validate the effectiveness and advantages of our approach across a range of challenging continuous control tasks.

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

## A    LIMITATIONS

One limitation we found is that our approach wasn't tested on discrete domains due to its incompatibility with discrete action spaces. Possible solutions could be using Gumbel-Softmax relaxation (Jang et al., 2017) or straight-through gradients (Bengio et al., 2013) for one-hot Categorical policy. Moreover, although the distributional Bellman operator is at best a non-expansion in KL divergence, we found it to be effective in practice. In the future, one question worth considering is whether $p(\mathcal{O} = 1|s, a)$ should be expanded under $p_\psi(U|s, a)$ or $q(U|s, a)$. For the latter, it is possible to decouple the policy evaluation from Equation 9 and utilize various existing methods for return distribution approximation.

## B    IMPLEMENTATION DETAILS

### B.1    MODEL ARCHITECTURE

We use the RSSM of (Hafner et al., 2020) and all other components as three dense layer of size 300 with ELU activation (Clevert et al., 2016). Both the policy and posterior are modeled as Beta distribution (Chou et al., 2017) due to its bounded support and analytical KL divergence. While both models share the same network architecture, investigating different model capacities is left for future research. The value distribution is modeled by a Normal distribution as suggested in Equation 11. The reward model is also represented by a Normal distribution. The posterior and policy are equipped with `LayerNorm` (Ba et al., 2016) for all layers while only the first layer for the value distribution. We use a planning horizon $H = 15$, and the number of trajectories is $N = 1$.

In addition, we add a noise $\mathcal{N}(0, \kappa^2)$ to the reward targets, where $\kappa$ is a constant. This choice is particularly beneficial for the sparse reward tasks, as the noise serves as a means of exploration.

Other aspects that distinguish DRIVE from Dreamer (Hafner et al., 2020) include: 1) we do not necessitate exploration noise during data collection, 2) we clip the gradient norm of the model to be below 150 instead of 100, and 3) we use $H$-step return rather than $\lambda$-return.

Our implementation is built on top of the open source code https://github.com/facebookresearch/denoised_mdp/tree/main.

### B.2    PSEUDOCODE

---

**Algorithm 2** DRIVE

---

**Denote** $x_t = (h_t, s_t)$
**Initialize parameters** $\phi, \theta, \psi$
**while** not converged **do**
    **for** each update step $c = 1, \ldots, C$ **do**
        ▷ Model Learning
        Sample $B$ sequences $\{(a_t, r_t, o_{t+1})\}_{t=k}^{k+L}$ of length $L$.
        Compute beliefs $h_t = \text{GRU}(h_{t-1}, s_{t-1}, a_{t-1})$.
        Compute posterior states $s_t \sim q(s_t|h_t, o_t)$.
        Update transition model (Equation 16).
        ▷ Behavior Learning
        Imagine $H$-length trajectories $\{(x_\tau, a_\tau)\}_{\tau=t}^{t+H}$ from each $x_t$ with $a_t \sim q_\phi(\cdot|\mathcal{O} = 1, x_t)$ otherwise $a_\tau \sim \pi_\theta(\cdot|x_\tau), \tau > t$.
        Sample rewards $r_{t+\tau} \sim p(r_{t+\tau}|x_{t+\tau}), \tau = 0, 1, \ldots, H - 1$.
        Sample values $U_{t+H} \sim p_\psi(U_{t+H}|x_{t+H}, a_{t+H})$.
        Compute $H$-step return as targets $\widehat{U}_t$ for each $(x_t, a_t)$.
        Estimate $q(U_t|x_t, a_t)$ with rewards $r_{t+\tau}$ and statistics $(\mu_{t+H}, \sigma_{t+H})$ (Equation 12).
        Update posterior and policy (Equation 14).
        Update value distribution with $\widehat{U}_t$ (Equation 13).
    **end for**
    ▷ Data Collection
    Initialize $h_0, s_0, a_0$.
    $o_1 \leftarrow$ env.reset().
    **for** each environment step $t = 1, \ldots, T$ **do**
        Compute the belief $h_t = \text{GRU}(h_{t-1}, s_{t-1}, a_{t-1})$.
        Compute the posterior state $s_t \sim q(s_t|h_t, o_t)$.
        Execute $a_t \sim \pi_\theta(\cdot|x_t)$.
        Observe reward $r_t$ and next observation $o_{t+1}$.
        Store transition $(a_t, r_t, o_{t+1})$ to the replay buffer $\mathcal{D}$.
    **end for**
**end while**

---

### B.3 HARDWARE

All our experiments were run on NVIDIA GeForce RTX 3090 with 24 GB memory. The rough execution time for each run is around 12h to finish 1M steps. We did not observe a significant difference in the computational complexity between DRIVE and Dreamer.

### B.4 HYPERPARAMETERS

| Name | Symbol | Value |
|---|---|---|
| **World Model** | | |
| Replay capacity (FIFO) | — | $10^6$ |
| Batch size | $B$ | 50 |
| Sequence length | $L$ | 50 |
| State size | — | 30 |
| Belief size | — | 200 |
| RSSM number of units | — | 200 |
| KL freenats | — | 3 |
| World model learning rate | — | $6 \cdot 10^{-4}$ |
| Model gradient clipping | — | 150 |
| **Behavior** | | |
| Imagination horizon | $H$ | 15 |
| Number of trajectories | $N$ | 1 |
| Discount | $\gamma$ | 0.99 |
| Actor learning rate | — | $8 \cdot 10^{-5}$ |
| Critic learning rate | — | $8 \cdot 10^{-5}$ |
| Actor gradient clipping | — | 100 |
| Critic gradient clipping | — | 100 |
| **Common** | | |
| MLP number of layers | — | 3 |
| MLP number of units | — | 300 |
| Action repeat | — | 2 |
| Adam epsilon | $\epsilon$ | $10^{-7}$ |
| Reward noise | $\kappa$ | sparse 0.3; dense 0.0 except 0.1 for `walker-stand` |
| **Others** | | |
| Random seeds | — | 0-4 |

Table 2: Hyperparameters of DRIVE.

## C EXPERIMENTAL DETAILS

### C.1 FIGURE 2

We examine the relationship between our optimality criterion and the transformed mean with respect to the return distribution. In the scatter plot at the top, for each policy update, we evaluate those two quantities on a batch of data. To approximate the expectation $\mathbb{E}_{p_\psi(U|s,a)}[\exp U]$, we sample 1000 return samples from $p_\psi(U|s,a)$ per data point, whereas for the transformed mean, we compute $Q(s,a) = \mathbb{E}_{p_\psi(U|s,a)}[U]$. Both measures are normalized by $\exp U_{\max}$ to ensure they lie within the range $[0,1]$. We plot our criterion against the transformed mean in ascending order, repeating this process periodically throughout training. In addition, we investigate how the variability within the return distribution influences the two criteria. For the plot at the bottom, we evaluate the correlation between the stddev of the return distribution and the two criteria on evenly spaced bins, each containing 100 samples from the batch. The data are also ordered by the transformed mean to ensure

that the correlation is calculated for samples with similar mean values, while allowing the stddev to vary.

## C.2 FIGURE 4

In Figure 4(a), we report the task mean along with the mean of 95% confidence intervals across 5 tasks: walker-walk, cheetah-run, quadruped-run, ball-in-cup-catch, and finger-spin. For the baselines, we either set the posterior equal to the policy, canceling the complexity term and the branching effect, or remove the regularizer term. In Figure 4(b), we report the aggregated performance on the walker-walk task while varying the number of trajectories $N$. Those trajectories are used to estimate the reparameterized PG $\mathcal{J}_U$ (Equation 10) and the regularizer term $\mathcal{J}_{\mathrm{KL}}^{(2)}$ (Equation 12).

## C.3 FIGURE 3

We consider a 6-armed truncated Normal bandit $\mathbf{T}(\mu_i, \sigma_i^2, m, M), 1 \le i \le 6$. We set $m = 1$ and $M = 10$. The remaining parameters for each arm $a_i$ are as follows:

- $a_1$: (1, 1)
- $a_2$: (1, 3)
- $a_3$: (5, 3)
- $a_4$: (10, 0.01)
- $a_5$: (10, 2)
- $a_6$: (9.9, 0.1)

Clearly, maximizing the expected value alone is insufficient to guarantee optimality, since variance also plays a vital role. Consequently, the standard definition of regret may no longer be appropriate:

$$\rho(T) = T\mu^\star - \sum_{t=1}^{T} \mu(a_t). \tag{27}$$

We adjust it by incorporating the variance, which emphasizes uncertainty when the expected value is high and reduces it otherwise:

$$\rho(T) = T\mu^\star - \sum_{t=1}^{T} \mu(a_t) + \sum_{t=1}^{T} \lambda(\mu(a_t))\sigma(a_t), \tag{28}$$

where $\lambda(\mu(a_t)) := \frac{\mu(a_t) - \min_i \mu(a_i)}{\max_i \mu(a_i) - \min_i \mu(a_i)}$. Under this criteria, the optimal action is $a_4$, as it has the highest expected value with high confidence. Although action $a_5$ attains the same mean, its higher variance makes it suboptimal. Furthermore, action $a_6$ is the second best action, even though it does not achieve the maximal expected value. For actions with mediocre expected values, high uncertainty might be preferred, as it offers the chance of achieving a higher value while, whereas actions with low uncertainty will never yield a high value. An uncertainty-agnostic policy, such as the greedy selection, does not take variance into account, therefore could easily become trapped in a suboptimal solution. In contrast, our method effectively balances exploration and exploitation, deciding when to explore and when to exploit.

## C.4 ADDITIONAL RESULTS

As a direct consequence of our probabilistic objective, the resulting policy is monitored through its entropy during training to investigate exploration at different stages (Figure 5(a)). We compare our method with DreamerV2, which explicitly includes an entropy term in the policy objective. We find that our policy exhibits higher entropy during the early stages of training and lower entropy at convergence, further supporting our claim about balancing exploration and exploitation. Additionally, we investigate the effect of different planning horizons on policy optimization (Figure 5(b)). We observe that, although the planning horizon does not significantly affect the average return near convergence, a longer horizon may lead to instability.

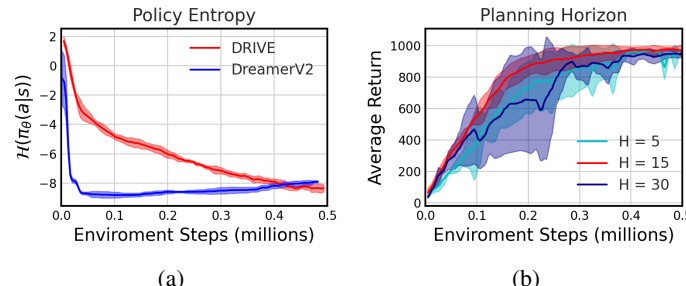

Figure 5: **(a)** Comparison of policy entropy; **(b)** Different horizons for model-based planning.

## D   DERIVATION OF VARIATIONAL BOUND

Note that:

$$p_\psi^{\pi_\theta}(\mathcal{O} = 1|s) = \int_a \pi_\theta(a|s)p_\psi(\mathcal{O} = 1|s,a)da, \tag{29}$$

and by using importance sampling and Jensen's inequality, henceforth we have,

$$
\begin{aligned}
\log p_\psi^{\pi_\theta}(\mathcal{O} = 1|s) &= \log \int_a \pi_\theta(a|s)p_\psi(\mathcal{O} = 1|s,a)da \\
&= \log \mathbb{E}_{a \sim q_\phi(a|\mathcal{O}=1,s)}\left[\frac{\pi_\theta(a|s)p_\psi(\mathcal{O}=1|s,a)}{q_\phi(a|\mathcal{O}=1,s)}\right] \\
&\geq \mathbb{E}_{a \sim q_\phi(a|\mathcal{O}=1,s)}\left[\log\frac{\pi_\theta(a|s)}{q_\phi(a|\mathcal{O}=1,s)} + \log p_\psi(\mathcal{O}=1|s,a)\right] \\
&= -D_{\text{KL}}(q_\phi(a|\mathcal{O}=1,s)||\pi_\theta(a|s)) + \mathbb{E}_{q_\phi(a|\mathcal{O}=1,s)}\left[\log p_\psi(\mathcal{O}=1|s,a)\right].
\end{aligned}
\tag{30}
$$

Next, we will expand $\log p_\psi(\mathcal{O} = 1|s,a)$ in similar procedures.

First of all, from our assumptions (1) $p(\mathcal{O} = 1|U, s, a) \propto \exp(U)$ and (2) $p(\mathcal{O} = 1|U_{\max}, s, a) = 1$, it is not difficult to tell that $p(\mathcal{O} = 1|U, s, a) = \frac{\exp(U)}{\exp(U_{\max})}$.

Then, with some algebra:

$$\log p_\psi(\mathcal{O} = 1|s,a) = \log \int p(\mathcal{O} = 1|U,s,a)p_\psi(U|s,a)dU \tag{31}$$

$$= \log \mathbb{E}_{\mathsf{q}(U|s,a)}\left[p(\mathcal{O} = 1|U,s,a)\frac{p_\psi(U|s,a)}{\mathsf{q}(U|s,a)}\right] \tag{32}$$

$$\geq \mathbb{E}_{\mathsf{q}(U|s,a)}[U] - D_{\text{KL}}(\mathsf{q}(U|s,a)||p_\psi(U|s,a)) - \text{const}, \tag{33}$$

where $\text{const} = U_{\max}$.

Finally, by plugging Equation 31 into Equation 30, the desired result is attained.

## E   DECOMPOSITION

**For $\mathcal{J}_{\text{KL}}^{(2)}$:**   From (Bellemare et al., 2017) and with an approximate transition model $\hat{f}$, we know that:

$$\mathsf{q}(U|s,a) = \frac{1}{\gamma^H}\mathbb{E}_{\pi_\theta,\hat{f}}\left[p_\psi(\frac{U - R_{<H}}{\gamma^H})\right]. \tag{34}$$

We further restrict value distribution to be Normal distribution, thus we have $p_\psi = \mathcal{N}(\mu_\psi, \sigma_\psi^2)$.

Then we can expand $\mathsf{q}(U|s,a)$ as follows:

$$\mathsf{q}(U|s,a) = \frac{1}{\gamma^H}\mathbb{E}_{\pi_\theta,\hat{f}}\left[p_\psi(\frac{U-R_{<H}}{\gamma^H})\right].$$

$$= \frac{1}{\gamma^H}\mathbb{E}_{\pi_\theta,\hat{f}}\left[\frac{1}{\sqrt{2\pi}\sigma_\psi(s_{t+H},a_{t+H})}\exp\left(-\frac{\left(\frac{U-R_{<H}}{\gamma^H}-\mu_\psi(s_{t+H},a_{t+H})\right)^2}{2\sigma_\psi^2(s_{t+H},a_{t+H})}\right)\right]$$

$$= \mathbb{E}_{\pi_\theta,\hat{f}}\left[\frac{1}{\sqrt{2\pi}(\gamma^H\sigma_\psi(s_{t+H},a_{t+H}))}\exp\left(-\frac{U-\left(R_{<H}+\gamma^H\mu_\psi(s_{t+H},a_{t+H})\right)}{2\left(\gamma^H\sigma_\psi(s_{t+H},a_{t+H})\right)^2}\right)\right]$$

$$= \mathbb{E}_{\pi_\theta,\hat{f}}\left[\mathcal{N}\left(R_{<H}+\gamma^H\mu_\psi(s_{t+H},a_{t+H}),\gamma^{2H}\sigma_\psi^2(s_{t+H},a_{t+H})\right)\right]$$

$$(35)$$

**For $\mathcal{J}_U$:** With:

(a) expand $\mathbb{E}_{\pi_\theta,\hat{f}}$ by definition.

(b) draw $\tau$-irrelevant variable $U$ inside the integral.

(c) change of variables, $z := \frac{U-R_{<H}}{\gamma^H}$.

(d) independence between return and history trajectory.

we have:

$$\mathcal{J}_U = \mathbb{E}_{q_\phi(a|\mathcal{O}=1,s),\mathsf{q}(U|s,a)}[U]$$

$$= \int_a q_\phi(a|\mathcal{O}=1,s)\int_U \mathsf{q}(U|s,a)U\,dU\,da$$

$$= \frac{1}{\gamma^H}\int_a q_\phi(a|\mathcal{O}=1,s)\int_U \mathbb{E}_{\pi_\theta,\hat{f}}\left[p_\psi(\frac{U-R_{<H}}{\gamma^H})\right]U\,dU\,da$$

$$\overset{(a)}{=} \frac{1}{\gamma^H}\int_a q_\phi(a|\mathcal{O}=1,s)\int_U\left[\int_\tau p(\tau|s,a)p_\psi(\frac{U-R_{<H}}{\gamma^H})d\tau\right]U\,dU\,da$$

$$\overset{(b)}{=} \frac{1}{\gamma^H}\int_a\int_U\int_\tau q_\phi(a|\mathcal{O}=1,s)p(\tau|s,a)p_\psi(\frac{U-R_{<H}}{\gamma^H})U\,d\tau\,dU\,da$$

$$\overset{(c)}{=} \frac{1}{\gamma^H}\int_a\int_z\int_\tau q_\phi(a|\mathcal{O}=1,s)p(\tau|s,a)p_\psi(z|\tau)(R_{<H}+\gamma^H z)d\tau(\gamma^H dz)da$$

$$\overset{(d)}{=} \int_a\int_z\int_\tau q_\phi(a|\mathcal{O}=1,s)p(\tau|s,a)p_\psi(z|s_{t+H},a_{t+H})(R_{<H}+\gamma^H z)d\tau\,dz\,da$$

$$= \mathbb{E}_{q_\phi,\pi_\theta,\hat{f},p_\psi(U|s_{t+H},a_{t+H})}[R_{<H}+\gamma^H U(s_{t+H},a_{t+H})]$$

$$(36)$$

# F  PROOFS

## F.1  PROOF OF THEOREM 5.1

*Proof.* To avoid the ambiguity when the corresponding terms are shorthanded, we denote:

$$p_U^\pi(s_{t+H},a_{t+H}) := p^\pi(U|s_{t+H},a_{t+H})$$
$$p_\mathcal{O}^\pi(s,a) := p^\pi(\mathcal{O}=1|s,a)$$

$$(37)$$

From Equation 18, if a change from $\pi$ to $\tilde{\pi}$ occurs, we know that:

$$\log p_H^\pi(\mathcal{O}=1|s,a;\tilde{\pi}) := \log\mathbb{E}_{\tilde{\pi},P,p_U^\pi(s_{t+H},a_{t+H})}\left[\exp\left(R_{<H}+\gamma^H U\right)\right] - U_{\max}, \quad (38)$$

When $\tilde{\pi} = \pi$, by definition of the value distribution, we further have:

$$\log p_H^\pi(\mathcal{O}=1|s,a,\pi) = \log p^\pi(\mathcal{O}=1|s,a). \quad (39)$$

Since the lefthand of Equation 38 is implicitly dependent on both $p_U^\pi$ and $\tilde{\pi}$, we will overload the notation $\mathcal{J}(q, \pi)$ to $\mathcal{J}(q, p_U^\pi, \pi)$.

We will start by inspecting the problem (a) where $\pi$ is fixed. Note that:

$$
\begin{aligned}
\mathcal{J}(q, p_U^\pi, \pi) &= -D_{\mathrm{KL}}(q||\pi) + \mathbb{E}_q[\log p_{\mathcal{O}}^\pi(s, a)] \\
&= \int_a q \log \frac{p_{\mathcal{O}}^\pi \pi}{q} da \\
&\stackrel{(a)}{=} \int_a q \log \frac{\frac{\exp{(Q^\pi)}\pi}{Z^\pi(s)}}{q} da + \log Z^\pi(s) \\
&= -D_{\mathrm{KL}}\left(q \middle|\middle| \frac{\exp{(Q^\pi)}\pi}{Z^\pi(s)}\right) + \log Z^\pi(s),
\end{aligned}
\tag{40}
$$

where (a) supplements the partition function $Z^\pi(s) = \mathbb{E}_\pi[p_{\mathcal{O}}^\pi]$ without changing the objective's quantity. This step ensures $\frac{p_{\mathcal{O}}^\pi \pi}{Z^\pi(s)}$ is a distribution.

Since the partition function only depends on $\pi$, it will have no effect of the optimization over $q$. Therefore, maximizing $\mathcal{J}(q, p_U^\pi, \pi)$ w.r.t. $q$ is equivalent to minimizing the KL divergence. It immediately follows that:

$$
q^\pi = \max_q \mathcal{J}(q, p_U^\pi, \pi) = \frac{p_{\mathcal{O}}^\pi \pi}{Z^\pi(s)}.
\tag{41}
$$

In addition, the above analysis guarantees the following relationship to hold:

$$
\mathcal{J}(q^\pi, p_U^\pi, \pi) \geq \mathcal{J}(q, p_U^\pi, \pi), \forall q.
\tag{42}
$$

Next, fixing $q^\pi$, we will try to solve the second-stage problem. For simplicity's sake, we replace $\log p_H^\pi(\mathcal{O} = 1|s, a; \tilde{\pi})$ with $\log p_H^\pi(\tilde{\pi})$. Then we try to optimize the following objective over $\tilde{\pi}$ with a fixed horizon $H$:

$$
\mathcal{J}(q^\pi, p_U^\pi, \tilde{\pi}) = -D_{\mathrm{KL}}(q||\tilde{\pi}) + \mathbb{E}_q[\log p_H^\pi(\tilde{\pi})].
\tag{43}
$$

We denote its maximizer as $\pi' = \arg\max_{\tilde{\pi}} \mathcal{J}(q, p_U^\pi, \pi)$. Then it must hold that:

$$
\mathcal{J}(q^\pi, p_U^\pi, \pi') \geq \mathcal{J}(q^\pi, p_U^\pi, \pi) \geq \mathcal{J}(q, p_U^\pi, \pi), \forall q.
\tag{44}
$$

The same logic would follow when it comes from $\pi$ to $\pi'$, that is:

$$
\mathcal{J}(q^{\pi'}, p_U^{\pi'}, \pi'') \geq \mathcal{J}(q^{\pi'}, p_U^{\pi'}, \pi') \geq \mathcal{J}(q, p_U^{\pi'}, \pi'), \forall q.
\tag{45}
$$

From the second inequality of Equation 45, it must hold for $q^\pi$ such that:

$$
\mathcal{J}(q^{\pi'}, p_U^{\pi'}, \pi') \geq \mathcal{J}(q^\pi, p_U^{\pi'}, \pi').
\tag{46}
$$

Due to the truncated optimization over finite horizon, how to bridge $\mathcal{J}(q^\pi, p_U^\pi, \pi')$ to $\mathcal{J}(q^\pi, p_U^{\pi'}, \pi')$ becomes a challenge. However, the condition 19 gives the tightest sufficient condition to ensure that:

$$
\begin{aligned}
\mathcal{J}(q^\pi, p_U^{\pi'}, \pi') - \mathcal{J}(q^\pi, p_U^\pi, \pi') &= -D_{\mathrm{KL}}(q^\pi||\pi') + \mathbb{E}_{q^\pi}\left[\log \mathbb{E}_{\tau|\pi', P, p_U^{\pi'}(s_{t+H}, a_{t+H})}\left[\exp{(R_{<H} + \gamma^H U)}\right]\right] \\
&\quad + D_{\mathrm{KL}}(q^\pi||\pi') - \mathbb{E}_{q^\pi}\left[\log \mathbb{E}_{\tau|\pi', P, p_U^\pi(s_{t+H}, a_{t+H})}\left[\exp{(R_{<H} + \gamma^H U)}\right]\right] \\
&= \mathbb{E}_{q^\pi}\left[\log \mathbb{E}_{\tau|\pi', P, p_U^{\pi'}(s_{t+H}, a_{t+H})}\left[\exp{(R_{<H} + \gamma^H U)}\right]\right] \\
&\quad - \mathbb{E}_{q^\pi}\left[\log \mathbb{E}_{\tau|\pi', P, p_U^\pi(s_{t+H}, a_{t+H})}\left[\exp{(R_{<H} + \gamma^H U)}\right]\right] \\
&= \mathbb{E}_{q^\pi}\left[\log \mathbb{E}_{\tau|\pi', P, p_U^{\pi'}(s_{t+H}, a_{t+H})}\left[\exp{(R_{<H} + \gamma^H U)}\right] \right.\\
&\quad \left. - \log \mathbb{E}_{\tau|\pi', P, p_U^\pi(s_{t+H}, a_{t+H})}\left[\exp{(R_{<H} + \gamma^H U)}\right]\right] \\
&= \mathbb{E}_{q^\pi}\left[\log \frac{\mathbb{E}_{\tau|\pi', P, p_U^{\pi'}(s_{t+H}, a_{t+H})}\left[\exp{(R_{<H} + \gamma^H U)}\right]}{\mathbb{E}_{\tau|\pi', P, p_U^\pi(s_{t+H}, a_{t+H})}\left[\exp{(R_{<H} + \gamma^H U)}\right]}\right] \\
&= \mathbb{E}_{q^\pi}\left[\log \frac{\mathbb{E}_{\tau|\pi', P}\left[\exp{(R_{<H})} g^{\pi'}(s_{t+H}, a_{t+H})\right]}{\mathbb{E}_{\tau|\pi', P}\left[\exp{(R_{<H})} g^\pi(s_{t+H}, a_{t+H})\right]}\right] \geq 0,
\end{aligned}
\tag{47}
$$

thereby leading to:

$$\mathcal{J}(q^\pi, p_U^{\pi'}, \pi') \geq \mathcal{J}(q^\pi, p_U^\pi, \pi') \tag{48}$$

Combining the relationships from Equation 44 and Equation 46, we have:

$$
\begin{aligned}
\log p^{\pi'}(\mathcal{O} = 1|s) &= \mathcal{J}(q^{\pi'}, p_U^{\pi'}, \pi') \\
&\geq \mathcal{J}(q^\pi, p_U^{\pi'}, \pi') \\
&\geq \mathcal{J}(q^\pi, p_U^\pi, \pi') \\
&\geq \mathcal{J}(q^\pi, p_U^\pi, \pi) \\
&= \log p^\pi(\mathcal{O} = 1|s)
\end{aligned}
\tag{49}
$$

Following this procedure, we can produce a sequence of $\log p^{\pi_k}(\mathcal{O} = 1|s), k = 0, 1, \cdots, \forall s \in \mathcal{S}$ that is monotonically increasing starting from a given initial policy $\pi_0$. Since we assume the reward function is bounded, the return distribution has a bounded support. Then by definition of $\log p^{\pi_k}(\mathcal{O} = 1|s)$, we know that it is also bounded. Therefore, the sequence converges to some $\pi^\star$ such that $\lim_{k\to\infty} \log p^{\pi_k}(\mathcal{O} = 1|s) = \log p^{\pi^\star}(\mathcal{O} = 1|s) = \sup_k \log p^{\pi_k}(\mathcal{O} = 1|s), \forall s \in \mathcal{S}$.

### F.1.1 RELATIONSHIP BETWEEN $\log p^{\pi^\star}(\mathcal{O} = 1|s)$ AND $V^{\pi^\star}(s)$

There are two questions we need to answer: **(1)** Given the local optimal policy $\pi^\star$ obtained by our proposed probabilistic learning proxy, what is the relationship between its corresponding value function? **(2)** Given a deterministic optimal policy $\pi^\star$ obtained by the value function, what is the relationship between its corresponding probabilistic learning proxy?

For the first question, note:

$$
\begin{aligned}
\log p^{\pi^\star}(\mathcal{O} = 1|s) &= \log \mathbb{E}_{\pi^\star}\left[ p^{\pi^\star}(\mathcal{O} = 1|s, a) \right] \\
&= \log \mathbb{E}_{\pi^\star, P, p^{\pi^\star}(U|s_{t+H}, a_{t+H})}\left[ \exp\left( R_{<H} + \gamma^H U \right) \right] - U_{\max} \\
&\geq \mathbb{E}_{\pi^\star, P, p^{\pi^\star}(U|s_{t+H}, a_{t+H})}\left[ R_{<H} + \gamma^H U \right] - U_{\max} \\
&= \mathbb{E}_{\pi^\star, P}\left[ R_{<H} + \gamma^H Q^{\pi^\star} \right] - U_{\max} \\
&= \mathbb{E}_{\pi^\star}\left[ Q^{\pi^\star} \right] - U_{\max} \\
&= V^{\pi^\star}(s) - U_{\max}.
\end{aligned}
\tag{50}
$$

For the second question, note:

$$\mathcal{L}(q) = -D_{\mathrm{KL}}(q||\pi^\star) + \mathbb{E}_{q, p^{\pi^\star}(U|s,a)}[U] - U_{\max}. \tag{51}$$

Using the fact that $Q^\pi(s, a) = \mathbb{E}_{p^\pi(U|s,a)}[U]$ for any $\pi$, we have:

$$\mathcal{L}(q) = -D_{\mathrm{KL}}(q||\pi^\star) + \mathbb{E}_q[Q^{\pi^\star}] - U_{\max}. \tag{52}$$

Since $\pi^\star$ is a deterministic optimal policy, therefore it is a Dirac delta distribution $\delta(a - a_0)$ upon some desired action $a_0$. By definition of the KL divergence, $q$ must be absolutely continuous with respect to $\pi^\star$ to have a finite value. Based on this, we know that:

$$
\mathcal{L}(q) = \begin{cases} V^{\pi^\star}(s) - U_{\max} & \text{if } q = \pi^\star \\ -\infty & \text{otherwise} \end{cases}
\tag{53}
$$

Therefore, it concludes that:

$$\max_q \mathcal{L}(q) = \mathcal{L}(\pi^\star) = V^{\pi^\star}(s) - U_{\max}. \tag{54}$$

$\square$

## F.2 PROOF OF THEOREM 5.2

*Proof.* Similarly, since $Q_H^\pi(s, a; \pi')$ defined in Equation 23 is implicitly dependent on both $Q^\pi$ and $\pi'$, we will overload the notation $\mathcal{L}(q, \pi)$ to $\mathcal{L}(q, Q^\pi, \pi)$.

For the first stage problem (a), the deduction is very similar, except we need to use the fact that $Q^\pi = \log \exp(Q^\pi)$.

$$
\begin{aligned}
\mathcal{L}(q, Q^\pi, \pi) &= -D_{\mathrm{KL}}(q||\pi) + \mathbb{E}_q[Q^\pi] \\
&\overset{(a)}{=} \int_a q \log \frac{\exp(Q^\pi)\pi}{q} da \\
&\overset{(b)}{=} \int_a q \log \frac{\frac{\exp(Q^\pi)\pi}{Z^\pi(s)}}{q} da + \log Z^\pi(s) \\
&= -D_{\mathrm{KL}}\left(q \Big|\Big| \frac{\exp(Q^\pi)\pi}{Z^\pi(s)}\right) + \log Z^\pi(s).
\end{aligned}
\tag{55}
$$

Then, the maximizer of $\mathcal{L}(q, Q^\pi, \pi)$ w.r.t. $q$ is

$$
q^\pi = \max_q \mathcal{L}(q, Q^\pi, \pi) = \frac{\exp(Q^\pi)\pi}{Z^\pi(s)}.
\tag{56}
$$

Henceforth, the following relationship holds:

$$
\mathcal{L}(q^\pi, Q^\pi, \pi) \geq \mathcal{L}(q, Q^\pi, \pi), \forall q.
\tag{57}
$$

Next, for the second-stage problem, we replace $Q_H^\pi(s, a; \pi')$ with $Q_H^\pi(\pi')$ beforehand.

Then, we optimize the following objective over $\tilde{\pi}$ with a fixed horizon $H$:

$$
\mathcal{L}(q^\pi, Q^\pi, \tilde{\pi}) = -D_{\mathrm{KL}}(q||\tilde{\pi}) + \mathbb{E}_q[Q_H^\pi(\pi')],
\tag{58}
$$

for which, the maximizer is $\pi' = \arg\max_{\tilde{\pi}} \mathcal{L}(q, Q^\pi, \pi)$. Then it must hold that:

$$
\mathcal{L}(q^\pi, Q^\pi, \pi') \geq \mathcal{L}(q^\pi, Q^\pi, \pi) \geq \mathcal{L}(q, Q^\pi, \pi), \forall q.
\tag{59}
$$

From the second inequality of Equation 59, it must hold for $\pi$ such that:

$$
\mathcal{L}(q^\pi, Q^\pi, \pi) \geq \mathcal{L}(\pi, Q^\pi, \pi).
\tag{60}
$$

Similarly, we can bridge $\mathcal{L}(q^\pi, Q^\pi, \pi')$ to $\mathcal{L}(q^\pi, Q^{\pi'}, \pi')$ with the condition 24 so that:

$$
\mathcal{L}(q^\pi, Q^{\pi'}, \pi') \geq \mathcal{L}(q^\pi, Q^\pi, \pi').
\tag{61}
$$

Furthermore, likewise in Equation 57, for the successor policy $\pi'$, we have:

$$
\mathcal{L}(q^{\pi'}, Q^{\pi'}, \pi') \geq \mathcal{L}(q, Q^{\pi'}, \pi'), \forall q.
\tag{62}
$$

Combining the relationships Equation 59, Equation 60, Equation 61, and 62, we have:

$$
\begin{aligned}
\log \mathbb{E}_{\pi'}[\exp Q^{\pi'}] &= \mathcal{L}(q^{\pi'}, Q^{\pi'}, \pi') \\
&\geq \mathcal{L}(q^\pi, Q^{\pi'}, \pi') \\
&\geq \mathcal{L}(q^\pi, Q^\pi, \pi') \\
&\geq \mathcal{L}(q^\pi, Q^\pi, \pi) \\
&= \log \mathbb{E}_\pi[\exp Q^\pi] \\
&\geq \mathcal{L}(\pi, Q^\pi, \pi) \\
&= V^\pi(s).
\end{aligned}
\tag{63}
$$

Following this procedure, we can produce a monotonically increasing bounded sequence of $\log \mathbb{E}_{\pi_k}[\exp Q^{\pi_k}], k = 0, 1, \cdots, \forall s \in \mathcal{S}$ starting from a given initial policy $\pi_0$. With similar deductions, the sequence converges to a local optimum $\pi^\star$ such that $\lim_{k \to \infty} \log \mathbb{E}_{\pi_k}[\exp Q^{\pi_k}] = \log \mathbb{E}_{\pi^\star}[\exp Q^{\pi^\star}] = \sup_k \log \mathbb{E}_{\pi_k}[\exp Q^{\pi_k}], \forall s \in \mathcal{S}$. □

