# OpenReview forum: "DRIVE: Distributional Model-Based Reinforcement Learning via Variational Inference"
_ICLR.cc/2025/Conference — Submitted to ICLR 2025_

### Official Review · Reviewer_EmnD · 2024-10-31

**Soundness:** 3
**Presentation:** 2
**Contribution:** 3
**Rating:** 6
**Confidence:** 3

**Summary:**

The authors propose a framework that leverages probabilistic inferencing to find locally-optimal policy from distributional value function. In particular, they propose a variational approach that, amongst other things, decouples the optimization problem from the regular value function. They provide a theoretical analysis in support of this framework, as well as empirical results in a broad range of continuous control tasks.

**Strengths:**

This paper provides a theoretical framework to find optimal policies in distribution RL, in which policy optimization does not fully utilize the distributional knowledge of value function in the existing literature. This is to complete the distributional approach to RL from end to end.

**Weaknesses:**

The ideas presented in the paper have potential, but overall, it lacks motivational discussion (or justification) on various components in the proposed framework: why model-based approach by address H-horizon, and Assumption 3.1. Also, discussion is missing on the impact of approximate transition model, the additional burden from model-based approach, impact of H.

The presentation of the experiments needs to be improved. In particular, by aggregating 11 experiments into a single figure (Fig. 2), we lose the ability to generate any meaningful insights as to which conditions may favor the author’s methods. Moreover, only some baselines are included in Figure 2, and soft-actor critic is excluded from Table 1, which is curious and not motivated in the text. Overall, there is no intuition provided by the authors as to why it makes sense to compare their method to the other methods, besides D4PG.

Finally, regarding the method proposed by the authors, I also have some concerns. First, the method appears to be much more computationally-expensive than, say, the more standard Distributional RL methods by Bellemare et al. (2017), due to the marginalization over the return, as well as the inclusion of Dreamer. Similarly, the inclusion of Dreamer would likely introduce some error into the learning processes, however this is not discussed or investigated by the authors.

**Questions:**

1.	Would Eq (4) be compatible with the Bellman Eq (2). Wouldn’t Eq (4) lead to entropy term in the reward?
2.	How do you define U_max in Assumption 3.1?
3.	The transition associated with the second inequality in Eq (9) is not clear to me.
4.	Where does Eq (13) come from?

---

> ### Author Response · Authors · 2024-11-25
>
> We would like to thank the reviewer for their thoughtful and constructive feedback. We have carefully considered the suggestions and address them in detail below.
> ## Weaknesses
> > impact of H
>
> We have provided new results in **Appendix C.4, Figure 5(b)** that investigate the effect of different horizons on model-based planning. We find that, although the planning horizon does not significantly affect the average return near convergence, a longer horizon may lead to instability.
> > The presentation of the experiments needs to be improved; motivation of the baselines
>
> We thank the reviewer for pointing out the presentation of experiments; we have therefore incorporated the full task-wise results in **Table 1**. In addition, since our algorithm is a distributional model-based RL algorithm under the "RL as inference" framework, we chose two model-based algorithms, two "RL as inference" algorithms, and one distributional RL algorithm as baselines to test the effectiveness of those aspects.
> > the method appears to be much more computationally-expensive
>
> Our algorithm does not perform full marginalization over the return; instead, we use the Monte carlo estimates to efficiently optimize the variational objective, as stated in *line 238*. For further investigation, we compare the **Second per Iteration** - the time required to complete one iteration of both model learning and behavior learning - against our base model, Dreamer. we do not observe significant overhead from the presence of the posterior network and the new objective, as shown in **Table 4(c)**.
> ## Questions
> > Q.1
>
> No, the Eq (2) is not compatible with Eq (4), as Eq (2) considers future rewards, whereas Eq (4) looks at the past rewards given a trajectory. For the second part, under a specific reformulation, it could lead to an entropy term, i.e. by assuming a uniform prior on the policy in Eq (2) as in [1] - but this is not commonly used.
> > Q.2
>
> The detailed derivation of $U_\text{max}$ can be found in *line 1004*.
> > Q.3
>
> Eq (9) can be derived by applying the Jensen's inequality and importance weighting using $\mathfrak{q}$.
> > Q.4
>
> Eq (13) comes from differentiating the variational objective in Eq (9) with respect to $\psi$.
> ## References
> [1] Abdolmaleki, A., Springenberg, J. T., Tassa, Y., Munos, R., Heess, N., & Riedmiller, M. (2018). Maximum a posteriori policy optimisation. arXiv preprint arXiv:1806.06920.

---

> ### Author Response · Authors · 2024-11-27
> **Seeking Your Feedback on Our Response**
>
> Dear Reviewer EmnD,
>
> Thank you for your time and effort in reviewing our work and providing valuable feedback. We would like to kindly ask if our responses have addressed all of your concerns. Your feedback is greatly appreciated.
>
> Once again, thank you for your consideration.
>
> Best regards,\
> Authors

---

> ### Author Response · Authors · 2024-11-28
> **A Gentle Reminder for Feedback**
>
> Dear Reviewer EmnD,
>
> We would like to thank you again for the time and effort you dedicated to reviewing our work. Your invaluable feedback has greatly contributed to its improvement. As the discussion period is nearing its end, we kindly request your thoughts on our comments and updates. We hope the additional experiments and clarifications have addressed your concerns. We would also appreciate knowing if there are any further aspects you would like us to address.
>
> If you have any additional concerns or questions, please let us know before the rebuttal period ends, and we will be happy to address them.
>
> Thank you once again for your time and consideration.
>
> Best regards,\
> Authors

---

> ### Author Response · Authors · 2024-11-29
> **A Friendly Reminder**
>
> Dear Reviewer EmnD,
>
> We sincerely thank you for your invaluable suggestions and feedback, which have helped us improve the quality of our work. We have followed your genuine suggestions, including:
>
> * Investigating of the impact of planning horizon in **Appendix C.4, Figure 5(b)**;
> * Improving the presentation of the experiments in **Table 1**;
> * Comparing the computational complexity in **Table 4(c)**.
>
> We have revised the paper accordingly since your last comment, and we believe these changes address all of your concerns. We would appreciate it if you could share your thoughts and comments on these updates.
>
> Thank you once again for your time and commitment.
>
> Best regards, \
> Authors

---

> ### Author Response · Authors · 2024-12-02
>
> Dear Reviewer EmnD,
>
> We greatly appreciated your questions and the opportunity to address them in our previous responses.
>
> With the discussion deadline approaching in one day, we would like to check if you have any additional feedback. Your input is valuable, and we would be happy to address any further points to ensure the work meets your expectations.
>
> Thank you for your time and thoughtful engagement. We look forward to hearing from you.
>
> Best regards, \
> Authors

---

> ### Author Response · Authors · 2024-12-03
> **Final Reminder**
>
> Dear Reviewer EmnD,
>
> We greatly appreciated your questions and the opportunity to address them in our previous responses.
>
> With the discussion deadline approaching in less than 12 hours, we would like to check if you have any additional feedback. Your input is valuable, and we would be happy to address any further points to ensure the work meets your expectations.
>
> Thank you for your time and thoughtful engagement. We look forward to hearing from you.
>
> Best regards, \
> Authors

---

### Official Review · Reviewer_6hNG · 2024-11-03

**Soundness:** 3
**Presentation:** 2
**Contribution:** 3
**Rating:** 8
**Confidence:** 3

**Summary:**

This paper proposes DRIVE, a novel distributional model-based reinforcement learning (RL) framework leveraging variational inference to address limitations in traditional RL approaches focused on expected return maximization. DRIVE introduces probabilistic learning proxies that incorporate aleatoric uncertainty within the return distribution, promoting risk-aware decision-making. The theoretical analysis demonstrates conditions for convergence, and empirical results on tasks from DMControl Suite highlight DRIVE's efficiency and its ability to balance exploration and exploitation.

**Strengths:**

Originality:
The integration of variational inference into distributional RL is novel, and facilitates a risk-aware decision-making approach

Significance:
The paper is well theoretically-founded and appears to be effective

Clarity:
The paper is well-written and logically structured

**Weaknesses:**

- Although the paper emphasizes balancing exploration and exploitation, it lacks a detailed analysis of how can DRIVE balance exploration and exploitation during traning, it would be helpful to include a detailed discussion.

- The multi-term objective and the complex posterior approximation might lead to overfitting, especially in limited-data scenarios. Specifically, how does your method scale to high-dimensional and large action-space environments

- Missing reference: The paper does not address a comparison with bisimulation mode based approaches [1, 2, 3]. How does your method relate to representation learning [1] and reward shaping methods [2,3] that also utilize the bisimulation metric for learning transition and reward models? These methods can be categorized as model-based approaches connected to the optimal value function. Specifically, [3] also manages to encourage exploration without loss of exploitation by connecting the shaping reward with the value function. What advantages does DRIVE offer over these bisimulation model-based approaches?

[1] Zhang et al. "Learning invariant representations for reinforcement learning without reconstruction." ICLR 2021

[2] Kemertas et al. "Towards robust bisimulation metric learning." NeurIPS 2021

[3] Wang et al. "Efficient potential-based exploration in reinforcement learning using inverse dynamic bisimulation metric." NeurIPS 2023

**Questions:**

- Can the authors provide more concrete evidence or metrics that demonstrate how their method manages the exploration-exploitation trade-off across different tasks?

- How does the complexity of the variational bound affect the scalability of the method in large state-action spaces?

- What measures, if any, have been implemented to mitigate overfitting due to the complex objective function and posterior approximation
Speficifically, how does DRIVE perform in discrete-action games, e.g., atari games?

- What advantages does DRIVE offer over these bisimulation model-based approaches[1,2]? (see W3)

- Some parts of the theoretical analysis are dense. Could you provide more intuitive explanations or examples to help bridge the gap between the theory and its practical implications?

---

> ### Author Response · Authors · 2024-11-25
>
> We thank the reviewer for their valuable feedback, which has helped improve the quality of our work. We address the reviewer's concerns in detail below.
> ## Weaknesses
> > it lacks a detailed analysis of how can DRIVE balance exploration and exploitation during traning,
>
> We have designed a diagnostic and minimal multi-armed bandit in the presence of aleatoric uncertainty to investigate how and why our method balances exploration and exploitation. We find that a greedy policy can easily become trapped in a suboptimal solution due to high variance, whereas our method effectively addresses this issue. Additionally, our method encourages exploration of actions with mediocre expected values but high uncertainty, as it offers the chance of achieving a higher value; in contrast, actions with low uncertainty will never yield a high value. For more details, please refer to **Figure 3** and **Appendix C.3**.
> > [...] might lead to overfitting; how does your method scale to high-dimensional and large action-space environments
>
> We do not find that our method suffers from issues of overfitting. Especially when paired with a sufficiently good model, we can query as much data as needed to alleviate this problem. For the second question, since our method works with high-dimensional images and action spaces, the current results highlight its effectiveness in this scenario.
> >  Missing reference on bisimulation mode based approaches
>
> It is worth noting that bisimulation-based approaches focus on **invariant representation learning**, whereas our method focuses on **policy search**. Therefore, these approaches should be considered complementary to ours, rather than something we aim to improve.
> ## Questions
> > more concrete evidence of balancing exploration and exploitation
>
> Please refer to our results from the multi-armed bandit experiment.
> > complexity of the variational bound; scalability
>
> Our variational objective can be efficiently optimized by stochastic gradient methods using Monte carlo estimates, so scalability is not a concern.
> > What measures, if any, have been implemented to mitigate overfitting; discrete-action games
>
> For the first part, since we do not observe any issues with overfitting, we don't have any tailored implementation. For the second part, this is a very good question. Unfortunately, since our joint optimization of the variational objective requires the reparameterization trick, it only applies to the continuous action spaces. One potential workaround for discrete action spaces would be to use Gumbel-Softmax relaxation [1] or straight-through gradients [2] for one-hot Categorical policy, which we leave for the future work.
> > practical implications of theory
>
> We have discussed why optimizing the lower bound is more practically feasible in *line 328-332*. Moreover, maximizing our objective is equivalent to maximizing an upper bound of the value function in standard RL.
> ## References
> [1] Jang, E., Gu, S., & Poole, B. (2016). Categorical reparameterization with gumbel-softmax. arXiv preprint arXiv:1611.01144.
>
> [2] Bengio, Y., Léonard, N., & Courville, A. (2013). Estimating or propagating gradients through stochastic neurons for conditional computation. arXiv preprint arXiv:1308.3432.

---

> > ### Comment · Reviewer_6hNG · 2024-11-28
> >
> > Thank you for your response and the additional experiment. I believe most of my concerns have been addressed. However, after thoroughly reviewing the technical details of Wang et al. [3], I would like to clarify that, as I mentioned earlier, [3] can be categorized as a model-based approach that learns both a reward model and transition models. The potential function is connected to the optimal value function, leveraging state-value information during exploration. It also keeps a balance between exploration and exploitation, which bears similarities to yours. Could you please elaborate more on how DRIVE's performance compares to the model-based reward shaping methods, particularly in scenarios where DRIVE learns more models than [3]? I am willing to increase my score if the question is answered satisfactorily.
> >
> >
> > [3] Wang et al. "Efficient potential-based exploration in reinforcement learning using inverse dynamic bisimulation metric." NeurIPS 2023

---

> ### Author Response · Authors · 2024-11-27
> **Seeking Your Feedback on Our Response**
>
> Dear Reviewer 6hNG,
>
> Thank you for your time and effort in reviewing our work and providing valuable feedback. We would like to kindly ask if our responses have addressed all of your concerns. Your feedback is greatly appreciated.
>
> Once again, thank you for your consideration.
>
> Best regards,\
> Authors

---

> > ### Author Response · Authors · 2024-11-28
> > **A Gentle Reminder for Feedback**
> >
> > Dear Reviewer 6hNG,
> >
> > We would like to thank you again for the time and effort you dedicated to reviewing our work. Your invaluable feedback has greatly contributed to its improvement. As the discussion period is nearing its end, we kindly request your thoughts on our comments and updates. We hope the additional experiments and clarifications have addressed your concerns. We would also appreciate knowing if there are any further aspects you would like us to address.
> >
> > If you have any additional concerns or questions, please let us know before the rebuttal period ends, and we will be happy to address them.
> >
> > Thank you once again for your time and consideration.
> >
> > Best regards,\
> > Authors

---

> ### Author Response · Authors · 2024-11-28
>
> Thank you for your response! After reviewing the details of the paper you mentioned, we believe our method differs from it in many aspects:
>
> * **Research Focus** Our work focuses on bridging "RL as inference" with distributional RL to create a new *policy objective*. In contrast, [1] focuses on combining reward shaping with the bisimulation metric under the traditional RL setting to design a new *intrinsic reward*. Thus, the two approaches involve different problem settings and pursue distinct goals.
>
> * **The Role of Models** Although both works utilize the transition model and reward model, the calculation of intrinsic reward (i.e. Inverse Dynamic Bisimulation Metric) in [1] heavily depends on the transition kernel and reward function. In contrast, our approach does not rely as strongly on these models, since our objective can in principle be approximated by samples from the environment, even without a model.
>
> * **Exploration** The exploration of our method is implicit, stemming from the policy objective itself. In contrast, the exploration in [1] is explicit, as it measures state discrepancy to generate a curiosity-driven intrinsic reward. Additionally, our method is adaptive in balancing exploration and exploitation, without requiring a hyperparameter to trade off extrinsic and intrinsic rewards, as seen in the author's implementation [2]. Moreover, our method is better suited for environments with aleatoric uncertainty, which is handled by the marginalization in Eq (7).
>
> * **Connection to** $V^{\star}(s)$ While the theoretical analysis in our paper demonstrates that our objective is an upper bound of the corresponding optimal value function, it does **not** indicate that we are maximizing the value function. Instead, we are optimizing the probabilistic learning proxy proposed in Eq (6), which leads to a different optimal policy. In contrast, the relationship shown in Theorem 4. of [1] preserves the policy invariance [3], ensuring that optimizing a transformed reward does not change the optimal policy.
>
> * **Model Complexity** Our method introduces only one additional posterior network, whereas [1] introduces two: the inverse dynamic and parameterized inverse dynamic bisimulation metric. Therefore, we do not learn more models than [1].
>
> We hope those clarifications address your concerns, please let us know if you have any further questions!
>
> [1] Wang, Y., Yang, M., Dong, R., Sun, B., & Liu, F. (2024). Efficient potential-based exploration in reinforcement learning using inverse dynamic bisimulation metric. Advances in Neural Information Processing Systems, 36.\
> [2] https://github.com/YimingWangMingle/liberty/tree/master \
> [3] Ng, A. Y., Harada, D., & Russell, S. (1999, June). Policy invariance under reward transformations: Theory and application to reward shaping. In Icml (Vol. 99, pp. 278-287).

---

> > ### Comment · Reviewer_6hNG · 2024-11-28
> >
> > Thank you for the quick reply. Most of my concerns have been addressed; I would still recommend adding a discussion of related model-based reward shaping works, such as [1] and Wang et.al, etc. I will increase my score accordingly.
> >
> > [1] Pathak et al. "Self-Supervised Exploration via Disagreement" ICML 2019

---

> > > ### Author Response · Authors · 2024-11-28
> > >
> > > Thank you! We will add a discussion in our later revision!

---

### Official Review · Reviewer_h7oS · 2024-11-04

**Soundness:** 2
**Presentation:** 2
**Contribution:** 3
**Rating:** 6
**Confidence:** 3

**Summary:**

The paper combines perspectives from RL as inference and distributional RL nicely through introducing the return random variable $U$, which offers a unique perspective as well as a new principled way to do policy search. The perspectives introduced in this paper can be helpful for future works that combine tools from RL, distributional RL, and probabilistic inference. Empirical results are reasonable.

**Strengths:**

**[Originality]**
The paper combines perspectives from RL as inference and distributional RL nicely through introducing the return random variable $U$, which offers a unique perspective as well as a new principled way to do policy search.

**[Quality]**
The idea is well motivated and empirical results are reasonable.

**[Clarity]**
The writing and derivations are for the most part easy to follow

**[Significance]**
The perspectives introduced in this paper can be helpful for future works that combine tools from RL, distributional RL, and probabilistic inference.

**Weaknesses:**

For me, the main weaknesses are incomplete empirical evaluations, and some (for me) confusing choices made in the variational lower bound derivations.

- In Figure 4, the 95 confidence interval (?) between ablations are overlapped, suggesting there is no statistically significant difference between their mean. More seeds can be helpful to separate out the differences.
- Taking more samples to approximate a distribution ($\mathfrak{q}$, see L248) actually results in *worse* performance (Figure 4b) is strange, and in my opinion the authors do not provide a sufficiently satisfactory explanation for this.
- It is not fully clear yet how the method’s behavior differs from existing work. The method seems to be roughly on-par with existing strong model-based methods (Figure 2), with potentially interesting differences in their exploration (Figure 3) that can be further investigated.
- I will ask about variational approximation in the “question” section below.

**Questions:**

1. I am confused by equation 29 (which impacts the lower bound derivation of equation 9). Equation 29 can be equivalently written as
$$\log p_\psi (\mathcal{O}=1 | s,a) = \log E_{p_\psi(U|s,a)} [\frac{\exp U}{ \exp U_{max}} ] \geq E_{p_\psi(U|s,a)}[U] - U_{max}$$
without the need to introduce distribution $\mathfrak{q}$. Since $p_\psi(U|s,a)$ is supposed to be the parametric return model (L183) which we can sample from (or even have in close form), why did the authors introduce an additional distribution $\mathfrak{q}$ (in Eq 30)?

2. Relating to above, $\mathfrak{q}$ seems to be used as the “true” return distribution (e.g. Eq 11), which makes it weird that it does not “need” to appear in the lower bound at all. Am I conceptually missing something here? It perhaps feels like $\log p^\pi_{\psi} (\mathcal{O}=1 | s)$ is not quite the correct quantity to lower-bound and optimize (but should instead be e.g. a quantity that does not depend on $\psi$)?

3. What is the parametric distribution $p_\psi(U|s,a)$, I presume it is a Gaussian?

4. For results: why are the per-game SAC and DreamerV3 results from Figure 2 not present in Table 1? Also, can the authors provide per-game performance for the 11 games for SAC and DreamerV3?

5. Clarity: for exploration, can the authors provide more details about the two criterias being compared in Figure 3, bottom (i.e. in equations)? Further, I do not feel Figure 3 top is sufficiently well-explained to make sense.

6. The effect of this new objective on exploration is very interesting, and I think worth investigating further. For instance by quantifying state coverage, and/or showing an illustrative exploration trace in simple environments? As an example, see Figure 2 and 7 of [Li 2024].

7. For clarity, can the authors provide equations of the objectives when “replacing the posterior with the policy and removing the regularizer term”? L451.

8. (L448) The word “disentanglement” has a lot of associated meanings in machine learning. Perhaps just “ablation” would suffice as the paragraph title.

9. Nit pick: please use “\citep” and \citet” properly. For eg citations on L40, L57 (amongst others) should only be in a single bracket, and citations such as Dabney et al on L488 should use “citet”.

[Li 2024] Li, Qiyang, et al. "Accelerating exploration with unlabeled prior data." Advances in Neural Information Processing Systems 36 (2024).

---

> ### Author Response · Authors · 2024-11-25
>
> We thank the reviewer for their insightful questions. We will address the reviewer's concerns in detail below.
>
> > The effect of this new objective on exploration is very interesting, and I think worth investigating further.
>
> We have designed a diagnostic and minimal multi-armed bandit in the presence of aleatoric uncertainty to investigate how and why our method balances exploration and exploitation. We find that a greedy policy can easily become trapped in a suboptimal solution due to high variance, whereas our method effectively addresses this issue. Additionally, our method encourages exploration of actions with mediocre expected values but high uncertainty, as it offers the chance of achieving a higher value; in contrast, actions with low uncertainty will never yield a high value. For more details, please refer to **Figure 3** and **Appendix C.3**.
> > Taking more samples to approximate distribution $\mathfrak{q}$ actually results in worse performance (Figure 4b) is strange.
>
> This is because more trajectories result in more samples, which scale to $O(B N H)$, where $B$ is batch size, $N$ is the number of generated trajectories, and $H$ is the horizon length. All those samples are fictitious and come from an *approximate model*. By performing some algebra with the default batch size and horizon length, but setting $N = 10$, we obtain approximately $\sim 375 k$ fictitious data, which could be unreliable due to model errors being exploited per policy update.
> > can the authors provide per-game performance for the 11 games for SAC and DreamerV3?
>
> Thank for your suggestion. We have incorporated the full task-wise results in **Table 1**.
> > Clarity on Figure 3: details about the two criteria; further explanation to the top
>
> The details of the two criteria can be found in **Appendix C.1**. Additionally, we have added some visualizations to Figure 3 to aid understanding.
> > another lower bound without distribution $\mathfrak{q}$
>
> There are **three** reasons. **First**, our objective can have a potentially tighter lower bound. Let's decompose the log-likelihood:
>
> $\log {p_{\psi}(\mathcal{O} = 1 | s, a)} = -D_{\text{KL}}(\mathfrak{q}(U |s, a) || p_{\psi}(U | s, a)) + \mathbb{E}_ {\mathfrak{q}(U | s, a)} [\log {p(\mathcal{O} = 1 | U, s, a)}] + D_{\text{KL}}(\mathfrak{q}(U | s, a) || p(U | s, a, \mathcal{O} = 1))$,
>
> The last term in which is the KL divergence between the distribution $\mathfrak{q}$ and another posterior over return. By maximizing the first two terms, we can get a tighter lower bound, as a result, a more accurate estimate of the log-likelihood. In contrast, the lower bound you provided has a constant Jensen gap, which may be a worse approximation to the true log-likelihood.
>
> **Second**, the algorithm is designed to consider future rewards, not only through the return distribution but also via the policy itself. The policy search becomes more effective when combined with a model, enhancing sample efficiency.
>
> **Lastly**, the lower bound you provided reduces to the mean of the return distribution, which corresponds to the Q function. This completely eliminates the uncertainty information of the return distribution, making it less effective for risk-aware decision-making.
> > Relating to above, $\mathfrak{q}$ seems to be used as the “true” return distribution
>
> I believe the reviewer may have a misunderstanding here. $\mathfrak{q}$ represents only $(\mathcal{T^{\pi}})^{H} U$, regardless of the quality of $p(U | s, a)$. Furthermore, the derivation for Eq 11 holds for any $p(U | s, a)$.
> > What is the parametric distribution $p_{\psi}(U | s, a)$, I presume it is a Gaussian?
>
> Yes, it is for simplicity. Other options could be Gaussian mixtures or linearly transformed Beta.
> > For clarity, can the authors provide equations of the objectives when “replacing the posterior with the policy and removing the regularizer term”?
>
> Simply set $q_{\phi}$ to $\pi_\theta$ in Eq (9).

---

> ### Author Response · Authors · 2024-11-27
> **Seeking Your Feedback on Our Response**
>
> Dear Reviewer h7oS,
>
> Thank you for your time and effort in reviewing our work and providing valuable feedback. We would like to kindly ask if our responses have addressed all of your concerns. Your feedback is greatly appreciated.
>
> Once again, thank you for your consideration.
>
> Best regards,\
> Authors

---

> > ### Author Response · Authors · 2024-11-28
> > **A Gentle Reminder for Feedback**
> >
> > Dear Reviewer h7oS,
> >
> > We would like to thank you again for the time and effort you dedicated to reviewing our work. Your invaluable feedback has greatly contributed to its improvement. As the discussion period is nearing its end, we kindly request your thoughts on our comments and updates. We hope the additional experiments and clarifications have addressed your concerns. We would also appreciate knowing if there are any further aspects you would like us to address.
> >
> > If you have any additional concerns or questions, please let us know before the rebuttal period ends, and we will be happy to address them.
> >
> > Thank you once again for your time and consideration.
> >
> > Best regards,\
> > Authors

---

> ### Author Response · Authors · 2024-11-29
> **A Friendly Reminder**
>
> Dear Reviewer h7oS,
>
> We sincerely thank you for your invaluable suggestions and feedback, which have helped improve the quality of our work. We have followed your thoughtful suggestions, including:
>
> * Providing further evidence of balancing exploration and exploitation in **Figure 3** and **Appendix C.3**;
> * Including the full task-wise results in **Table 1**;
> * Adding visualizations and explanations in **Figure 3**.
>
> Additionally, we have made clarifications on various aspects in the comments, including:
>
> * The choice of variational objective;
> * The increasing model error harming the performance;
> * The misunderstanding of $\mathfrak{q}$;
> * The parameterization of the return distribution;
> * The details of removing the posterior.
>
> We have revised the paper accordingly and answered your questions in detail since your last comment. We believe these changes address most of your concerns. We would appreciate it if you could share your thoughts and comments on these updates.
>
> Thank you once again for your time and commitment.
>
> Best regards, \
> Authors

---

> ### Comment · Reviewer_h7oS · 2024-11-30
>
> Many thanks to the authors for their response, for adding the additional experiments, and various clarifications. Before I finalize my score, I hope to clarify a few more things which I (perhaps naively) did not fully understand yet.
>
> ## More clarification about the lowerbound discussion
>
> As I understand it, we want to lowerbound $\log p^\pi(\mathcal{O}=1 | s, a)$ (Eq 7, with derivation in L1003). It seems like there are at least two ways to do so.
>
> **Option A** (one I mentioned), with $E_p$ meaning expectation over $p_\psi (U|s,a)$:
>
> $$\log p^\pi(\mathcal{O}=1 | s, a) = \log E_p [ \frac{\exp U}{\exp U_m} ] \geq E_p[U] - U_{m}$$
>
> **Option B** (what this paper does), with bootstrapped return distribution $q (U|s,a)$:
> $$\log p^\pi(\mathcal{O}=1|s,a) = \log E_q [p^\pi(\mathcal{O}=1 | U,s,a) \frac{p(U|s,a)}{q(U|s,a)}] $$
>
> $$\geq E_q [U] - U_m - D_{KL} (q(U|s,a) || p_\psi (U|s,a))$$
>
> Assuming the authors agree with the above (please let me know if I made a mistake somewhere), the question now is which one is a tighter lower bound. The answer seems unclear to me. One the one hand, Option B has a non-negative KL term making the bound looser, and in the case that $p_\psi = q$ it would be identical. On the other hand, estimating $\mathbb{E}[U]$ from $q(U|s,a)$ could "somehow" be better than $p_\psi (U|s,a)$. I'm very curious if the authors have some intuition or thoughts here.
>
> The authors also mention that
> > By maximizing [option B], we can get a tighter lower bound. [Option A] has a constant Jensen gap.
>
> I am confused for how Option B can be optimized: the only parameter present seems to be $\psi$ which just minimize the KL term, no?
>
> > Second, the algorithm is designed to consider future rewards, not only through the return distribution but also via the policy itself. The policy search becomes more effective when combined with a model, enhancing sample efficiency.
>
> I don't really understand what the authors are trying to say here. What does "consider future rewards through the policy itself" mean? What is the "model" referring to here? The Dreamer world model? Or $q(U|s,a)$? If the former, one could also estimate $\mathbb{E}_{p (U|s,a)} [U]$ (from Option A above) via sampling the Dreamer model imagination, therefore making (real environment) sample efficiency a non-concern, no? Or do the authors mean in-imagination sample efficiency?
>
> > Lastly, the lower bound you provided reduces to the mean of the return distribution, which corresponds to the Q function. This completely eliminates the uncertainty information of the return distribution, making it less effective for risk-aware decision-making.
>
> This is an interesting point I'd like to appreciate further. If the authors agree with the derivations of Options A and B above, we see they differ mainly in $D_{KL}\left(q(U|s,a) || p_\psi(U|s,a)\right)$, with the only other term being an expectation of $U$, right? So is it the KL term that contains uncertainty information? Or $q(U|s,a)$ has uncertainty info while $p_\psi(U|s,a)$ doesn't? Please try to explain this to me simply -- I'm really trying to get an intuition for if/why this works.
>
> I'm also curious to square the above with the statement (L182 and equation 7):
> > By marginalizing over $U$, we can incorporate [aleatoric] uncertainty...
>
> Do authors mean Option A doesn't incorporate aleatoric uncertainty? Both Options A and B marginalize over $U$, no?
>
> ---
>
> I'd like to emphasize that none of the above questions are rhetorical, and I'm not trying to say "Option A is better" or "authors should do Option A". It does not matter to me which option is used as long as it is well-motivated, and that any limitations are clearly stated.
>
> I am currently genuinely confused: based on the paper's flow, the authors seem to make decisions in the derivation to turn "easy" problems into more complicated problems before solving the more complicated problems, when the "easy" problems seems directly solvable to me. I'd like to believe these decisions are well-motivated, but the benefit of the complexity isn't fully clear to me yet (at least based on the current paper's presentation). I'd really appreciate if the authors can try to explain this simply to me, if possible.

---

> ### Author Response · Authors · 2024-11-30
>
> Thank you for your further questions. We will address them in detail below.
> > Tightness of the lower bound
>
> First of all, the tightness of the lower bound cannot simply be attributed to the non-negative KL term; instead, it depends on both $\mathfrak{q}(U | s, a)$ and $p_{\psi}(U | s, a)$. Only when a sufficiently amount of number of data are given (which is rarely feasible), can $p_{\psi}(U | s, a)$ attain the true return distribution $p^{\pi}(U | s,a)$, so that the two options converge to the same objective. Otherwise, we need to account the discrepancy between $p_{\psi}(U | s, a)$ and $\mathfrak{q}(U | s, a)$ as a measure of the quality of return distribution for the decision-making process. The larger the discrepancy, the more unreliable the return distribution, and thus the worse the policy objective approximation. This is one practical benefit of our objective: being divergence-aware, reflected via the variational posterior $q_{\phi}(a | \mathcal{O} = 1, s)$ to the policy. On the other hand, regarding the tightness of the lower bound, we acknowledge you made a valid point that $q$ plays a vital role. In fact, the log-likelihood can be decomposed as follows:
>
> $\log {p_{\psi}(\mathcal{O} = 1 | s, a)} = -D_{\text{KL}}(\mathfrak{q}(U |s, a) || p_{\psi}(U | s, a)) + \mathbb{E}_ {\mathfrak{q}(U | s, a)} [\log {p(\mathcal{O} = 1 | U, s, a)}] + D_{\text{KL}}(\mathfrak{q}(U | s, a) || p(U | s, a, \mathcal{O} = 1))$
>
> By maximizing the first two terms, we are minimizing the KL divergence to another posterior $p(U | s, a, \mathcal{O} = 1)$. When $\mathfrak{q} = p(U | s, a, \mathcal{O} = 1)$, we obtain the exact log-likelihood. Now, the only question is *what drives $\mathfrak{q}$ towards $p(U | s, a, \mathcal{O} = 1)$*? The answer is that $\mathfrak{q}$ has a parametric dependence on the policy $\pi_\theta$, so the policy will be *partially* adjusted to push $\mathfrak{q}(U | s, a; \pi_\theta)$ closer to $p(U | s, a, \mathcal{O} = 1)$, resulting in a potentially tighter lower bound. The distribution $\mathfrak{q}(U | s, a; \pi_\theta)$ can be viewed as a semi-variational distribution, as it has separate functions to directly optimize the policy in Eq (10) and distill the knowledge from the variational posterior $q_{\phi}(a | \mathcal{O} = 1, s)$. Those are also what we implemented in practice.
> > Incorporating the future rewards into the policy objective itself
>
> When using $\mathbb{E}_{p(U | s, a)}[U]$, it can *not* use the Dreamer imagination, as it doesn't have a compact connection to the multi-step return, as shown in Eq (10). Intuitively, only the bootstrapped return distribution, $\mathfrak{q}$, can look into the future, whereas $p(U | s, a)$ can only resort to immediate samples from the return distribution. This is analogous to the difference between the value function $V(s)$ and the $n$-step bootstrapped value function $(\mathcal{T}^{\pi})^{(n)} V(s)$, where $\mathcal{T}^{\pi}$ is the standard Bellman expectation operator.
> > Source of aleatoric uncertainty
>
> The aleatoric uncertainty comes from marginalizing the return distribution with the utility function $p(\mathcal{O} = 1 | U, s, a) \propto \exp (U)$. Intuitively, this approach considers the tail events by squashing low values while incentivizing high ones, thus incorporating the information about the variability of the distribution. In contrast, the linear utility function does not capture this variability; it merely reflects the central tendency of the distribution rather than its spread. To illustrate, consider a concrete example $X \sim \mathcal{N}(\mu, \sigma^2)$, the linear utility function simply gives $\mathbb{E}[X] = \mu$, whereas the exponential utility function yields $\mathbb{E}[\exp (X)] = \exp (\mu + \sigma^2 / 2)$, which includes the variance term. Based on this, we can conclude that option A should be considered the incorrect objective, as it entirely disregards the uncertainty information of the return distribution and has a constant Jensen gap to the true objective. In contrast, option B mitigates this by using the semi-variational distribution $\mathfrak{q}$ as mentioned before.
>
> We hope those clarifications address your concerns regarding the variational objective. To summarize:
>
> * Option A is an incorrect objective with respect to aleatoric uncertainty.
> * Option B handles this by using the semi-variational distribution $\mathfrak{q}$.
> * Option B also enables both long-horizon and divergence-aware planning.
>
> We look forward to your response. Thank you again for your time and attention.

---

> > ### Comment · Reviewer_h7oS · 2024-12-01
> >
> > I thank the authors for the detailed explaintions, the motivations of the choices are now more clear to me.
> >
> > I personally would encourage the authors to try to further improve the clarity of the paper. A few example improvements (as they surfaced throughout our discussion) could include:
> > - Make more explicit in text the connection between components of Equation 9 to standard RL commponents (for instance, the analogy of $p_\psi (U | s, a)$ and $q(U | s, a)$ to the value function and n-step bootstrapped return). Maybe this can be added to the paragraph after Eq 9.
> > - The discussion such as why $q(U | s, a)$ may be closer to $p(U | s, a, \mathcal{O}=1)$ than $p_\psi (U | s, a)$ was very helpful for me and could be added to the text or appendix as well. In general, some confusion may arise when jumping between _RL concepts_ and _probabilistic inference concepts_. With my probabilistic inference hat, it is unclear why we need both $p_\psi$ and $q$, although with my RL hat, using the bootstrapped return is almost "obvious" -- relying on the RL-specific notion of "semi-gradients" (I believe $p_\psi(U|s,a)$ is used to estimate $q(U|s,a)$).
> >
> > I think this paper's primary contribution lies is its theoretical contributions (which I think is very interesting), rather than the currently presented empirical results. Therefore, I believe making clearer and more intuitive things such as the connection of DRIVE with "standard" RL approaches, the motivations behind the derivation for DRIVE, and benefits of the DRIVE objective (Eq 9) over standard ones could greatly increase the impact of this paper in the space of idea for RL researchers (even if these things are obvious to the authors).
> >
> > All in all, I think my concerns about the complexity of the derivation have been largely addressed and I've raised my score accordingly.

---

> > > ### Author Response · Authors · 2024-12-01
> > >
> > > We sincerely thank the reviewer for their insightful questions and suggestions throughout the discussion. We also appreciate the reviewer's acknowledgement of our contributions. We will integrate the suggestions you emphasized in the next revision, including:
> > >
> > > * A discussion of the connection between return distributions and their value counterparts;
> > > * A discussion of the role of $\mathfrak{q}(U | s, a)$.
> > >
> > > We believe those suggestions will make the core concepts of the paper clearer, improve overall clarity, and highlight the benefits of the proposed objective.
> > >
> > > Once again, we thank the reviewer for their time and attention.

---

### Official Review · Reviewer_m9WT · 2024-11-05

**Soundness:** 3
**Presentation:** 2
**Contribution:** 2
**Rating:** 3
**Confidence:** 5

**Summary:**

This paper proposes a combination of distributional RL and probabilitstic inference in model based RL to incorporate aleatoric uncertaity of distribution for risk-aware decision-making.

The main learning objecitve of this paper is the log-likelihood of the optimality variable that is computed by marginalizing the conditional probability $p(O=1|U,s,a)$ on $U$ where $U$ is the return distribution.

Since the optimality variable $O=1$ implies the optimal return at given state and action, maximizing the return of agent induces the log-likelihood of optimality variable.

Compared to Dreamer, the above log-likelihood is added to the learning objective.

In experiments, all experiments is conducted in vision-based deepmind control suite.

**Strengths:**

The main contribution of this paper is to reduce the effect of aleotoric uncertainty in training model-based RL.

By introducing the optimality variable, the proposed method can asses the quality of return distribution by imagination and match to the ground-truth return distribution better by preventing from leaning to the maximum value.

**Weaknesses:**

The main conern that I have is the quality of presentation and the lack of experiments.

1. The notations are not consistent. For example, there are several different q functions which outputs  $a$ in Equation 9, $s_t$ in Equation 16, $U$ in Algorithm 1 without any proper subscriptions.
2. The theoretical results, theorem 5.1 and theorm 5.2 are not connected to the main contribution which induces the better performance of model-based distributional RL method.
3. It is not clear which distributinoal RL method is used to estimate $U$, such as C51 or QRDQN.
4. The experiments is only conducted in deepmind control suite, which is not reported in the main baseline, Dreamer. If the proposed method has an advantage in vision-based continuous control, the authors should have described how to perform better in continous control tasks.

**Questions:**

Can you make the notations be clear to help to understand the main contribution better?

---

> ### Author Response · Authors · 2024-11-25
>
> We thank the reviewer for their helpful feedback. We address the reviewer's concerns in detail below.
> > The notations are not consistent
>
> First and foremost, it is worth noting that we have utilized different font styles for different q functions in Eq (9) to avoid any ambiguity, which was not raised by the other reviewers. Regarding other existing notation issues, we have revised the font style for the representation model notation in both Eq (15) and (16), which we believe to be minor, and added subscripts for $U$ in **Algorithm 1** to enhance clarity and prevent any potential confusion.
> > practical connection of theoretical results
>
> Our theoretical analysis provides insights into why optimizing the lower bound is more practically feasible (see *lines 328-332*). Moreover, maximizing our objective is equivalent to maximizing an upper bound of the value function in standard RL, which connects to the traditional setting. Most importantly, it characterizes the optimization process, offering an explicit objective and guiding the approximate scenario.
> > It is not clear which distributinoal RL method is used to estimate $U$
>
> We neither use C51 nor QRDQN. Instead, our objective for return distribution is derived by differentiating the variational objective in Eq (9) with respect to $\psi$, which yields the cross-entropy loss. In practice, it is optimized using Monte carlo estimates.
> > advantage over baseline Dreamer
>
> The results in **Table 1** show that our method on the DeepMind Control Suite outperforms Dreamer by a wide margin.
>
> Overall, we respectfully encourage the reviewer to revisit the key aspects of the paper. With improved clarity, we believe the main contributions are now clearer and easier to understand.

---

> > ### Comment · Reviewer_m9WT · 2024-11-28
> >
> > Thank you for your response. While I acknowledge some errors in my initial review, my main concerns have not been adequately addressed. Let me provide more detailed comments about these concerns.
> >
> > 1. The authors can use the subscript for the notations on $q$ for the better readability.
> > 2. As far as I understand, Theorem 5.1 and 5.2 show policy improvement and the connectino with standard RL, respectively. In abastract, the author metioned that the proposed method incorporates aleatoric uncertainty of the return distribution, enabling risk-aware decision-making, but there is no specific mathematical analysis on this part.
> > 3. I think that distributional RL method should have the specific distribution representation regardless of considering return as distribution. However, the notation $U$ itself is just return distribution, not a learnable value distribution function. In addition, the corss-entropy loss can be used in the categorical or discrete setting, but $U$ is a continous value to represent the discounted sum of rewards. I think that the author intends to mention the log-likelihood, considering the definition in Eq. (13).
> > 4. It is not also clear that which policy (critic) is used in this paper for continuous control.
> > 5. For the result part, I find the result on DMControl suite for the baseline in their original paper. However, there are 22 environments in DMContol suite, but the authors have only reported the result on 12 envs.

---

> ### Author Response · Authors · 2024-11-27
> **Seeking Your Feedback on Our Response**
>
> Dear Reviewer m9WT,
>
> Thank you for your time and effort in reviewing our work and providing valuable feedback. We would like to kindly ask if our responses have addressed all of your concerns. Your feedback is greatly appreciated.
>
> Once again, thank you for your consideration.
>
> Best regards,\
> Authors

---

> ### Author Response · Authors · 2024-11-28
> **A Gentle Reminder for Feedback**
>
> Dear Reviewer m9WT,
>
> We would like to thank you again for the time and effort you dedicated to reviewing our work. Your invaluable feedback has greatly contributed to its improvement. As the discussion period is nearing its end, we kindly request your thoughts on our comments and updates. We hope the additional experiments and clarifications have addressed your concerns and will allow for a more favorable consideration. We would also appreciate knowing if there are any further aspects you would like us to address.
>
> If you have any additional concerns or questions, please let us know before the rebuttal period ends, and we will be happy to address them.
>
> Thank you once again for your time and consideration.
>
> Best regards,\
> Authors

---

> ### Author Response · Authors · 2024-11-28
>
> Thank you for your valuable feedback! We address your concerns in details below.
> > Question 1
>
> We appreciate the reviewer's suggestion to use subscripts for the q functions to improve readability. We will consider this in our next revision.
> > Question 2
>
> It is worth noting that, as indicated in both abstract (*line 025*) and the beginning of section 5 (*line 295-301*), the goal of the theoretical analysis is to detail the conditions for convergence, understand the optimization process, and establish a connection with standard RL. We believe those aspects provide equally valuable insights into the practical objective. Regarding the risk-aware decision-making, we have provided detailed experimental analyses to investigate how and why our method balances exploration and exploitation. Overall, our contributions lie in both the theoretical analysis and the experimental verification, which are complementary to each other.
> > Question 3
>
> **Firstly**, we aim to make our method as general as possible; therefore, we do not restrict the return distribution to specific classes. **Secondly**, it appears there may be a misunderstanding here, $U$ itself is a random variable, *not* a distribution; its distribution, $p(U | s, a)$, becomes learnable when parameterized. **Lastly**, the cross-entropy loss in Eq. (13) can indeed be used with categorical representations, either by using fixed bins as in C51 [1] or a set of quantiles as in QRDQN [2]. However, $U$ does not have to be continuous, which depends on the environment.
> > Question 4
>
> Both the policy and the posterior are involved in continuous control and are updated alternatively. However, only the policy is used for evaluation.
> > Question 5
>
> Although we acknowledge the reviewer's concern regarding the number of tested tasks, due to computational constraints, we were unfortunately unable to conduct all the experiments. However, we included representative tasks from both sparse and dense reward settings, with varying action dimensions, to make our evaluation as comprehensive as possible. These tasks are also used by the strong model-based RL algorithm [3]. Notably, our method outperforms Dreamer by a wide margin in the current task set, and we expect this improvement to hold across a broader variety of tasks.
>
> We hope those clarifications address your concerns and will allow for a more favorable consideration. Thank you!
>
> [1] Bellemare, M. G., Dabney, W., & Munos, R. (2017, July). A distributional perspective on reinforcement learning. In International conference on machine learning (pp. 449-458). PMLR. \
> [2] Dabney, W., Rowland, M., Bellemare, M., & Munos, R. (2018, April). Distributional reinforcement learning with quantile regression. In Proceedings of the AAAI conference on artificial intelligence (Vol. 32, No. 1). \
> [3] Hansen, N., Wang, X., & Su, H. (2022). Temporal difference learning for model predictive control. arXiv preprint arXiv:2203.04955.

---

> ### Author Response · Authors · 2024-11-29
> **Does Our Response Address Your Concerns?**
>
> Dear Reviewer m9WT,
>
> We sincerely thank you for the detailed and valuable feedback you provided earlier. We would like to kindly ask whether our latest responses have addressed all of your concerns. Your feedback is greatly appreciated.
>
> We understand that you may be busy due to the heavy workload of rebuttals and discussions. However, we hope you can reconsider your initial review, particularly since you acknowledged some errors. Additionally, it seems to contrast with the feedback from the other reviewers.
>
> If there are any further questions or aspects you would like us to address, please let us know! We would be happy to assist.
>
> Thank you for your time and commitment.
>
> Best regards,\
> Authors

---

> ### Author Response · Authors · 2024-12-01
> **A Friendly Reminder**
>
> Dear Reviewer m9WT,
>
> We sincerely thank you for your invaluable suggestions and feedback. As the discussion period is nearing its end, we kindly ask if our previous responses have addressed your concerns.
>
> We are pleased to have received positive feedback from the other three reviewers and are eager to ensure that our response has adequately addressed your concerns as well.
>
> It is important to note that our work integrates both theoretical and empirical studies, each offering crucial perspectives. Only when viewed holistically can the full scope of the contributions be appreciated. We therefore encourage the reviewer to reconsider their initial assessment.
>
> If there are any further questions or aspects you would like us to address, we would be happy to assist.
>
> Thank you again for your time and commitment.
>
> Best regards, \
> Authors

---

> ### Author Response · Authors · 2024-12-02
>
> Dear Reviewer m9WT,
>
> We greatly appreciated your questions and the opportunity to address them in our previous responses.
>
> With the discussion deadline approaching in one day, we would like to check if you have any additional feedback. Your input is valuable, and we would be happy to address any further points to ensure the work meets your expectations.
>
> Thank you for your time and thoughtful engagement. We look forward to hearing from you.
>
> Best regards, \
> Authors

---

> ### Author Response · Authors · 2024-12-03
> **Final Reminder**
>
> Dear Reviewer m9WT,
>
> We greatly appreciated your questions and the opportunity to address them in our previous responses.
>
> With the discussion deadline approaching in less than 12 hours, we would like to check if you have any additional feedback. Your input is valuable, and we would be happy to address any further points to ensure the work meets your expectations.
>
> Thank you for your time and thoughtful engagement. We look forward to hearing from you.
>
> Best regards, \
> Authors

---

### Author Response · Authors · 2024-11-26
**Summary of Changes**

We greatly appreciate the constructive comments from the reviewers. In response, we have revised the paper based on their suggestions. The updates are summarized as follows:

* We provided further evidence of balancing exploration and exploitation by designing a minimal, diagnostic multi-armed bandit experiment at the request of **`Reviewer h7oS`** and **`Reviewer 6hNG`**.
* We added a comparison of second per iteration in **Table 4(c)** to address the computational concern raised by **`Reviewer EmnD`**.
* We provided the full task-wise results for all baselines in **Table 1** at the request of **`Reviewer h7oS`** and **`Reviewer EmnD`**.
* We added a visualization and textual explanation in Figure 2 to aid understanding.
* We revised the font style for the q function in both Eq (15) and (16), and added subscripts for $U$ in **Algorithm 1** to enhance clarity as pointed out by **`Reviewer m9WT`**.
* We provided additional experimental results in **Appendix C.4**: one monitoring the policy entropy and the other investigating the impact of the planning horizon, as requested by **`Reviewer EmnD`**.

We have highlighted all the revised/added parts in **dark blue** for the convenience of the reviewers. We hope those changes address the reviewers' concerns, and we look forward to receiving their feedback.

---

### Meta-Review · Area_Chair_qac8 · 2024-12-21

**Metareview:**

The paper proposes a new model-based distributional RL technique based on variational inference.  It claims to enable risk sensitive decision making.  However this claim is not supported by suitable theoretical or empirical evidence.  The paper observes that previous techniques in control as inference define the probability of optimality to be proportional to the expected return of a trajectory.  Similarly, the paper proposes to use the exponential of the sampled return of a trajectory as a surrogate of optimality.  This is an interesting idea, but it lacks a clear mathematical derivation that demonstrates how risk sensitive decision making is achieved.  In fact, since the paper proposes to define the probability of optimality in terms of exponential returns, it seems that the literature on transformations of returns applies here.  More specifically, transforming returns with a concave function enables risk averse decision making while transforming returns with a convex function enables risk seeking decision making.  Since the paper proposes to take the exponential of the returns, it seems to enable risk seeking decision making only (not risk averse decision making).  See https://en.wikipedia.org/wiki/Risk_aversion for a discussion.  This needs to be clarified.

Ultimately, the theory presented simply analyzes the convergence of the algorithm, but does not show that the resulting policy can be risk averse nor how to adjust the degree of risk aversion.  Similarly, the experiments do not clearly demonstrate how to obtain risk averse policies nor how to adjust the degree of risk aversion.

While the paper investigates an interesting construction, there is a lack of justification and evidence for the proposed construction.  Hence this work is not ready for publication.

**Additional Comments On Reviewer Discussion:**

While three reviews are positive, the discussion of the reviewers focused on the lack of justification and evidence to confirm that the proposed construction can enable risk sensitive decision making as claimed.  A mathematical derivation is missing to confirm how risk aversion sensitive decision making is achieved and how the degree of risk can be adjusted.  The experiments do not provide suitable evidence either.  The area chair personally read the paper and shares those concerns too.

---

### Decision · Program_Chairs · 2025-01-22

Reject